# Scaling Direct Feedback Learning with Jacobian Alignment Guarantees

**Paul Caillon[1], Erwan Fagnou[1], Blaise Delattre[1] & Alexandre Allauzen[1,2]**
[1] Miles Team, LAMSADE, Université Paris Dauphine-PSL, Paris, France
[2] ESPCI PSL, Paris, France
`{name}.{surname}@dauphine.psl.eu`

## Abstract

Deep neural networks rely on backpropagation (BP) for optimization, but its strictly sequential backward pass hinders parallelism and scalability. Direct Feedback Alignment (DFA) has been proposed as a promising approach for parallel learning of deep neural networks, relying on fixed random projections to enable layer-wise parallel updates, but fails on deep convolutional networks, and performs poorly on modern transformer architectures. We introduce GrAPE (Gradient-Aligned Projected Error), a hybrid feedback-alignment method that (i) estimates rank-1 Jacobians via forward-mode JVPs and (ii) aligns each layer's feedback matrix by minimizing a local cosine-alignment loss. To curb drift in very deep models, GrAPE performs infrequent BP anchor steps on a single mini-batch, preserving mostly parallel updates. We show that the forward-gradient estimator has strictly positive expected cosine with the true Jacobian. We relate this estimator-level guarantee to a standard stochastic-approximation result under a positive expected-cosine condition on the update direction, providing theoretical support for GrAPE's alignment objective. Empirically, GrAPE consistently outperforms prior alternatives to BP, enabling the training of modern architectures, closing a large fraction of the gap to BP while retaining layer-parallel updates for the vast majority of steps.

## 1 Introduction

Backpropagation (BP) (Rumelhart et al., 1986) remains the de facto standard for training deep networks. However, its memory footprint and energy cost have become critical bottlenecks as architectures deepen and scale up. In particular, two properties of the BP impede the development of parallel training methods: the weight symmetry between the forward and backward pass, and the sequential propagation of the error. These two properties also clash with biological plausibility. In this paper, we deliberately set aside the biological concerns to focus on non-sequential alternatives to BP. A rich literature has mainly explored two independent axes of relaxation:

**Randomized feedback** (Feedback Alignment, FA (Lillicrap et al., 2016), Direct FA, DFA (Nøkland, 2016), etc.) replaces transposed weights by fixed or adaptive random matrices, but suffers from misalignment on deep or convolutional layers (Bartunov et al., 2018; Moskovitz et al., 2018; Launay et al., 2019). Adaptive variants using weight mirroring (Akrout et al., 2019) can approach BP performance, remaining sequential, however, offering limited practical advantages.

**Forward-gradient and forward-only** methods (Silver et al., 2021; Baydin et al., 2022; Hinton, 2022; Dellaferrera & Kreiman, 2022) replace the backward pass by Jacobian-vector products or a second "perturbed" forward pass, at the cost of high variance and limited scaling to modern architectures.

Our work extends the first axis, starting from the following observation: fixed random feedback matrices often lose positive cosine similarity with true gradients in deep and structured layers (Nøkland, 2016; Refinetti et al., 2021). As a consequence, this kind of feedback fails to decrease the loss function. We therefore introduce a lightweight and data-driven correction using forward gradient estimates. With this alignment, we combine the efficiency of randomized feedback with estimator-level Jacobian alignment guarantees from forward-gradient estimates, augmented by an occasional BP calibration step to reduce variance in very deep networks. Our core contributions are:

1. *Gradient-guided feedback.* We introduce **GrAPE** (Gradient Aligned Projected Error), which computes a local cosine-alignment loss with forward-gradient estimates. This realigns each layer's feedback matrix toward Jacobian-aligned directions prior to the parallel DFA update.

2. Leveraging forward-mode gradients, we derive a positive expected alignment bound for our rank-1 Jacobian estimator. We also recall a standard conditional convergence-in-expectation result under a positive expected-cosine assumption on the update direction, which provides theoretical motivation for GrAPE's alignment objective.

3. *Occasional BP calibration.* To further mitigate drift in very deep or highly structured networks, we apply a true BP step to a single mini-batch every $T$ epochs, using its exact gradient to realign the weights. This yields a hybrid two-timescale scheme in which most updates are layer-parallel GrAPE steps, interleaved with sparse BP synchronizations.

4. *Scalability.* We show for the first time that a DFA-style method can train VGG-16, ResNet-20/56 and Transformer models, narrowing the performance gap with full BP.

The paper is organized as follows: in Section 2 we briefly recall the necessary background and notation (a more detailed survey can be found in the Appendix). Section 3 describes the GrAPE algorithm and the occasional BP calibration strategy. Section 4 reports empirical results.

## 2 BACKGROUND AND RELATED WORKS

Let $f(x;\theta)$ be a feed-forward neural network with $L$ layers, where $x \equiv h_0$ is the input and $\theta = \{W_l\}_{l=1}^{L}$ is the set of parameters. Each layer computes $a_l = W_l h_{l-1}$ followed by a non-linearity $h_l = \sigma_l(a_l)$, encompassing both linear and convolutional operations. The output is $\hat{y} = h_L$. Given a loss function $\mathcal{L}(\hat{y}, y)$, the goal of backpropagation (BP) is to compute gradients $\nabla\mathcal{L}_l = \partial\mathcal{L}/\partial a_l$ recursively, starting from the output layer. The corresponding weight update is:

$$\delta W_l = \begin{cases} -\eta\,\nabla\mathcal{L}_L\, h_{L-1}^\top & \text{if } l = L \\ -\eta\,\delta a_l\, h_{l-1}^\top & \text{if } l < L, \end{cases} \quad \text{with} \quad \delta a_l = (W_{l+1}^\top \delta a_{l+1}) \odot \sigma_l'(a_l) \tag{1}$$

This algorithm is by construction sequential: the update at layer $l$ depends on the backpropagation of errors through all subsequent layers. This reliance on weight symmetry and stepwise computation hinders parallelism. As architectures attain increasing size and depth, alternative methods that allow non-symmetric error transmission and enable parallelized training have emerged (see Figure 1).

### 2.1 LEARNING WITH RANDOM FEEDBACK

**Feedback Alignment (FA)** proposes a biologically inspired alternative to backpropagation by replacing transposed weights with fixed random feedback matrices $B_l$ (Lillicrap et al., 2016). The error is still propagated sequentially, but independently of the forward weights ($W_l$):

$$\delta a_l = (B_l \delta a_{l+1}) \odot \sigma_l'(a_l), \quad \text{with } \delta a_L = (B_L \nabla\mathcal{L}_L) \odot \sigma_L'(a_L)$$

This removes the weight symmetry constraint, aligning better with biological learning (Lillicrap et al., 2020) but fails to scale to convolutional networks (Bartunov et al., 2018; Moskovitz et al., 2018). Adaptive variants using weight mirroring (Akrout et al., 2019) can however approach BP performance, but remain sequential and thus offer limited practical advantages.

**Direct Feedback Alignment (DFA)** (Nøkland, 2016) removes the need for sequential error propagation by projecting the output error directly to each hidden layer:

$$\delta a_l = (B_l \nabla\mathcal{L}_L) \odot \sigma_l'(a_l), \quad \forall l \in [1, L] \tag{2}$$

This enables parallel updates but remains limited on complex architectures like CNNs and Transformers. Attempts to mitigate this include adaptive feedback (e.g., weight mirroring (Akrout et al., 2019)) or architectural variants like DRTP (Frenkel et al., 2021), falling short behind BP on large-scale tasks.

Launay et al. (2020) applied DFA to Transformers using either block-wise ('macro') or layer-wise ('micro') feedback, yet BP remains necessary within attention layers. Our method builds on this approach by providing more informative feedback signals, complementing the internal backpropagation still required within attention blocks.

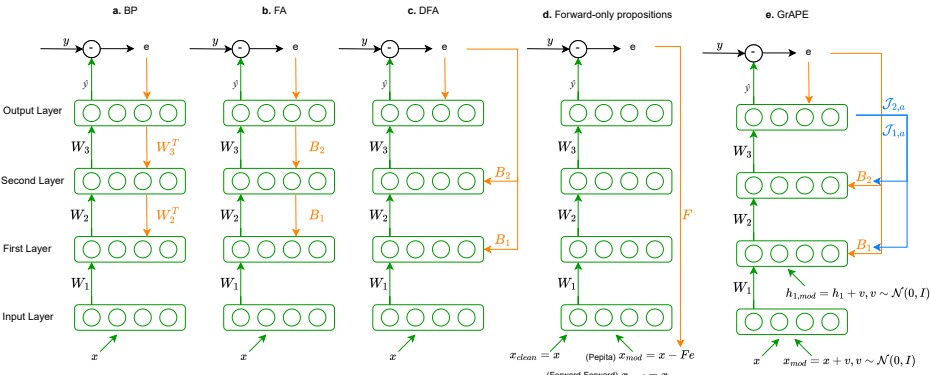

Figure 1: Overview of error propagation schemes, adapted from Dellaferrera & Kreiman (2022). (a) BP, (b) FA, (c) DFA, (d) forward-only methods (PEPITA, Forward-Forward), (e) GrAPE. Green arrows indicate forward paths; orange ones error signals and blue ones forward gradient estimates $\mathcal{J}_{i,a}$. Learned weights are denoted $W_l$, and layer-specific feedback weights as $B_l$.

The effectiveness of FA and DFA relies on the alignment between feedback vectors and true gradients as shown by Nøkland (2016); Refinetti et al. (2021). A sufficient condition for descent is thus: $\forall l \in [1, L], \quad \cos(\omega_l) = \frac{\nabla \mathcal{L}_l^\top B_l}{\|\nabla \mathcal{L}_l\| \cdot \|B_l\|} > 0$. This reflects the classical Zoutendijk condition (Nocedal & Wright, 1999), highlighting the importance of alignment for descent and convergence in classical deterministic line-search methods (see Appendix A.3).

## 2.2 FORWARD-ONLY CALCULATIONS

**Forward-only** methods aim to completely bypass the backward pass. For instance in PEPITA (Dellaferrera & Kreiman, 2022) or Forward-Forward (Hinton, 2022), a *double forward pass* provides a surrogate error signal via a perturbed forward step: $\delta W_l = (h_l - h_l^{err}) \odot (h_{l-1}^{err\ \top}), \quad h_0^{err} = x - Fe$

**Forward Gradient (FG)** methods (Silver et al., 2021; Baydin et al., 2022) use forward-mode automatic differentiation (FwAD) to obtain unbiased gradient estimates via directional (Jacobian–vector) derivatives along a random direction $\mathbf{u}$, removing the need for an explicit backward pass: $\nabla \mathcal{L} \cdot \mathbf{u} = \lim_{\delta \to 0} \frac{\mathcal{L}(\theta + \delta \mathbf{u}) - \mathcal{L}(\theta)}{\delta}$. While unbiased in theory, sampling in parameter space is inefficient for large models. Ren et al. (2022) address this by perturbing neuron activations instead of weights, significantly reducing variance and cost as activation space is usually much smaller than weight space. Additional improvements use local auxiliary losses (Fournier et al., 2023), requiring however BP to train the auxiliary models.
FwAD methods parallel the standard forward pass and can be implemented via dual numbers. Despite some runtime overhead (43% w.r.t. a simple forward with a naive implementation (Baydin et al., 2022)), optimized frameworks (e.g., PyTorch FwAD, JAX) promise broader applicability.

Other recent local learning rules also exploit forward computations without relying on FA/DFA-style feedback pathways. Nøkland & Eidnes (2019) attach shallow local classifiers and similarity-based objectives to each hidden layer, computing layerwise gradients from these auxiliary losses in a strictly feedforward manner (with an optional FA-based variant), while Apolinario et al. (2025) propose a rule that drives each layer to align its activations with fixed periodic basis vectors using a local cross-entropy loss. In both cases, learning signals are generated from local forward computations and do not rely on separate feedback matrices $B_l$ or on backpropagated errors from the global task loss.

While these methods can reach reasonable performance on relatively shallow or compact architectures, their reported accuracies typically remain below backpropagation on deeper models, and they have not yet been demonstrated at scale on modern Transformer-style networks. GrAPE is complementary: it also leverages forward-mode information, but uses JVP-based rank-1 Jacobian estimates to explicitly learn feedback matrices that approximate true gradient directions for DFA-style updates.

## 3 GRADIENT ALIGNED PROJECTED ERROR (GRAPE)

Here, we introduce GrAPE (Gradient-Aligned Projected Error). This method leverages the lightweight forward-gradient estimates to align the feedback matrices with the true Jacobians. This alignment is motivated by the Zoutendijk theorem, enabling scalable training of complex architectures on various tasks. We summarize the overall procedure in Algorithm 1.

### 3.1 LIMITATIONS OF FIXED RANDOM FEEDBACK MATRICES

In line-search methods, the update direction must align as closely as possible with the true negative gradient. DFA uses a fixed "feedback" direction in place of $-\nabla\mathcal{L}$, but cannot measure or correct the angle between its update and the actual descent direction. In convolutional layers, for example, the linear transformation can be represented by a block-Toeplitz matrix (d'Ascoli et al., 2019). Reproducing such a structure with a single, fixed, randomly sampled feedback matrix is impossible, as discussed by Refinetti et al. (2021). This explains why Launay et al. (2019) found that vanilla DFA often fails on convolutional networks: the convolutional weights cannot correctly capture the projected error if the feedback direction is misaligned with the true gradient. However, if we *align* the feedback projections with the associated gradients, this limitation can be overcome.

Beyond fixed random feedback, early weight-mirroring schemes (Akrout et al., 2019) already showed that it is possible in principle to adapt feedback weights to approximate $W_l^\top$ in FA, albeit in a fundamentally sequential setting tied to the forward weights.

Subsequent DFA-style approaches (Webster et al., 2021; Bacho & Chu, 2024) extend this idea, either by using a Kolen–Pollack–type learning rule to adapt feedback weights or by tracking BP updates through auxiliary forward passes and momentum. Roy et al. (2025) "unlock" SVD-space by optimizing a composite set of 5 local losses, including a cosine-like term, to align feedback with forward singular vectors. While these methods can substantially improve feedback alignment and sometimes recover BP-level accuracy, they either inherit the sequential nature of FA, rely on complex SVD-based or multi-term loss machinery, or, in the case of FDFA, raise reproducibility questions as discussed in Appendix A.4.

None of them exploit forward-mode JVPs to obtain a simple, analytically tractable Jacobian alignment guarantee for rank-1 estimators. GrAPE takes a different route: it uses JVP-based rank-1 Jacobian estimates and a single cosine loss to learn feedback matrices that are both amenable to layer-parallel DFA-style execution and analytically linked to the true Jacobians through a positive expected Frobenius cosine bound at the estimator level. Together with infrequent BP steps on a single mini-batch, GrAPE demonstrates in our experiments scaling to deeper CNNs and Transformer architectures where vanilla DFA and related methods have historically struggled.

### 3.2 ALIGNMENT WITH FORWARD GRADIENTS: STATISTICAL GUARANTEES

**Alignment lower bound** Consider the Jacobian matrix $\mathcal{J}_l = \frac{\partial \hat{y}}{\partial \mathbf{h}_l} \in \mathbb{R}^{d_{\text{out}} \times n_l}$, the perturbation $\mathbf{p} \sim \mathcal{N}(0, I_{n_l})$ and the Jacobian–vector product $\mathcal{J}_l\,\mathbf{p} \in \mathbb{R}^{d_{\text{out}}}$. An *unbiased* rank-1 approximation of $\mathcal{J}_l$ is then $\widehat{\mathcal{J}_l} = (\mathcal{J}_l\,\mathbf{p})\,\mathbf{p}^\top \in \mathbb{R}^{d_{\text{out}} \times n_l}$. We measure alignment via the Frobenius cosine

$$\cos_F(\mathcal{J}_l, \widehat{\mathcal{J}_l}) = \frac{\langle \mathcal{J}_l, \widehat{\mathcal{J}_l}\rangle_F}{\|\mathcal{J}_l\|_F\,\|\widehat{\mathcal{J}_l}\|_F}, \qquad \langle A, B\rangle_F := \text{Tr}(A^\top B). \tag{3}$$

Because $\mathbf{p} \sim \mathcal{N}(0, I_{n_l})$, we can write $\mathbf{p} = r\,\mathbf{s}$ with $\mathbf{s}$ uniform on the unit sphere and independent of $r = \|\mathbf{p}\|$. A direct computation shows $\cos_F(\mathcal{J}_l, \widehat{\mathcal{J}_l}) = \frac{\|\mathcal{J}_l\mathbf{s}\|}{\|\mathcal{J}_l\|_F}$, so $\mathbb{E}\Big[\cos_F(\mathcal{J}_l, \widehat{\mathcal{J}_l})\Big] = \frac{1}{\|\mathcal{J}_l\|_F}\,\mathbb{E}_{\mathbf{s}}\|\mathcal{J}_l\mathbf{s}\|$. Projecting onto the top singular direction of $\mathcal{J}_l$ and using a standard bound on the first coordinate of a uniform sphere point (Appendix B) yields

$$\mathbb{E}\Big[\cos_F(\mathcal{J}_l, \widehat{\mathcal{J}_l})\Big] \geq \sqrt{\frac{2}{\pi\,n_l}}\,\frac{\|\mathcal{J}_l\|_2}{\|\mathcal{J}_l\|_F}, \tag{4}$$

which is strictly positive for $\mathcal{J}_l \neq 0$. For the batched estimator (average of $B$ independent rank-1 estimates), Gaussian concentration implies that the empirical Frobenius cosine concentrates around equation 4 at rate $O(1/\sqrt{B})$ (Appendix B).

Beyond this layerwise alignment bound, Appendix B.4 recalls a standard stochastic-approximation result (Theorem B.1) showing that, under usual step-size conditions, a positive expected cosine between the update direction and the true gradient is sufficient for convergence to stationarity in expectation. This provides a theoretical support for learning feedback matrices that improve layerwise alignment. GrAPE is designed so that its JVP-based estimator and alignment loss encourage improved layerwise alignment, consistent with such a positive expected-cosine condition.

**Zoutendijk theorem** In classical smooth deterministic optimization, the Zoutendijk theorem states that if each search direction forms an angle uniformly bounded away from $\pi/2$ with the negative gradient and the step sizes satisfy Goldstein or strong Wolfe conditions (typically enforced by line-search; see Nocedal & Wright, 1999), then the gradient norm converges to zero and the iterates approach a stationary point. In our context, we view Zoutendijk primarily as a *conceptual lens* motivating cosine-based alignment: our JVP-based estimator yields a strictly positive expected Frobenius cosine between $\mathcal{J}_l$ and $\widehat{\mathcal{J}}_l$ (Eq. 4), and standard stochastic-approximation results (Theorem B.1 in Appendix B.4) show that a positive expected cosine between the update direction and the true gradient is sufficient for convergence to stationarity in expectation under usual step-size conditions. This motivates the alignment objective used to train the feedback matrices.

Concretely, we update each feedback matrix with respect to the corresponding estimated $\widehat{\mathcal{J}}_l$, before applying the DFA weight update (Equation 2). Here, the per-layer cosine with the estimated Jacobian serves as a tractable local alignment objective that heuristically supports global descent when improved across layers:

$$\forall l \in [1, L], \cos(\overline{\omega}_l) := \cos_F\big(B_l, \widehat{\mathcal{J}}_l\big) = \frac{\langle B_l, \widehat{\mathcal{J}}_l \rangle_F}{\|B_l\|_F \, \|\widehat{\mathcal{J}}_l\|_F} > 0, \tag{5}$$

In Appendix B.5, we further show via a simple Frobenius cosine composition lemma (Lemma B.2) that quantitative lower bounds on $\cos_F(B_\ell, \widehat{\mathcal{J}}_\ell)$ and $\cos_F(\widehat{\mathcal{J}}_\ell, \mathcal{J}_\ell)$ induce a corresponding lower bound on $\cos_F(B_\ell, \mathcal{J}_\ell)$. This controls how estimator noise and imperfect feedback learning combine, although positivity of the two intermediate cosines alone does not automatically imply a positive composed bound. In practice, we use the empirical average of per-column cosines $\bar{c}_l$ as a computationally convenient proxy for the Frobenius cosine $\cos_F(B_l, \widehat{\mathcal{J}}_l)$: as detailed in Appendix C, $\cos_F$ is a weighted average of these columnwise cosines, and after column normalization of $B_l$ and our JVP construction, the weights are close to uniform, so $\bar{c}_l$ is a convenient scalar summary of layerwise alignment.

In practice, for each layer we can choose to do a perturbation either in the weight space (as explained above) or in the activation space (Ren et al., 2022). Since the variance and cosine of the estimator depend directly on the dimension of the perturbations, we pick the space with the lowest dimension. Although usually the weight space is of much higher dimension than the activity space, this is not the case for the first layers of a deep convolutional network, for instance.

### 3.3 LEARNING RULE AND ALGORITHM

We first define a local *alignment loss*: $\mathcal{L}_{\text{align}}(B_l) = 1 - \cos(\overline{\omega}_l)$, where $\cos(\overline{\omega}_l)$ is defined in Equation 5. We update $B_l$ by one gradient step on $\mathcal{L}_{\text{align}}$:

$$B_l \leftarrow B_l - \eta_{B_l} \nabla_{B_l} \mathcal{L}_{\text{align}}(B_l), \quad \eta_{B_l} > 0, \tag{6}$$

and then normalize columns to enforce purely directional alignment, $B_l[:,k] \leftarrow B_l[:,k]/(\|B_l[:,k]\| + \varepsilon)$ with $\varepsilon > 0$ for numerical stability (cf. Nøkland, 2016).

This local step uses only forward-mode JVPs and no BP is required. It is worth noting that a single batched JVP per layer adds roughly a forward-pass-like cost and does not scale with the number of parameters. Furthermore, our rank-1 Jacobian estimate has a strictly positive expected Frobenius cosine with the true Jacobian and concentrates as the batch size grows (Appendix B). This guarantee concerns estimator–Jacobian alignment and provides theoretical support for the full GrAPE update. We found one alignment step per batch sufficient in practice; additional steps offered no gain.

Finally, with this refined $B_l$ we perform the parallel update:

$$\delta a_l = \big(B_l \nabla \mathcal{L}_L\big) \odot \sigma'_l(a_l), \qquad \delta W_l = -\eta \, \delta a_l \, h_{l-1}^\top.$$

### 3.4 SPARSE BP CALIBRATION ON A SINGLE MINI BATCH

In order to counteract the increased variance of our forward-gradient estimates in high dimensions, we additionally inject a true BP step on a single mini-batch, using its exact gradients to re-anchor all $W_l$ with minimal interruption of the parallel update flow. Let $T$ denote the number of epochs between two such BP calibration steps. Currently, we select the calibration mini batch uniformly at random; a promising extension would be to apply active-learning strategies to pick the most informative examples for each BP calibration step, for example with uncertainty sampling or core-set selection (Settles, 2009; Sener & Savarese, 2017).

The amortized per-epoch overhead of a single-mini-batch calibration is $\approx 1/(TN_b)$ of an epoch in units of mini-batches ($\mathcal{O}(N_b + 1/T)$ vs. $\mathcal{O}(N_b)$ steps). For typical $N_b \gg 1$, this is small in practice. Appendix D provides a compact FLOPs and critical-path accounting of BP, DFA, and GrAPE—covering forward+JVP overhead, per-layer projection/alignment, and the $\frac{1}{T}$ calibration term—together with preliminary timing results on a small transformer. A full study of optimized parallel kernels is left to future work.

---

**Algorithm 1** GrAPE

---

**Require:** Layers $1, \dots, L$, weights $\{W_l\}$, feedbacks $\{B_l\}$, BP-interval $T$, epochs $E$, batch size $B$, learning rates $\eta, \eta_B$
 1: **Forward pass & JVPs:** choose perturbation $p_l$ in the smaller of activation/weight space; run a single forward trace carrying duals
 2: **for** epoch = 1 to $E$ **do**
 3:     **for all** minibatch $(X, Y)$ of size $B$ **do**
 4:         **1) Forward pass & JVPs:** choose perturbation $p_l$ in smaller of activation/weight space
 5:         $h_0 \leftarrow X$
 6:         **for** $l = 1$ to $L$ **do**
 7:             $a_l \leftarrow W_l h_{l-1}, \quad h_l \leftarrow \sigma_l(a_l), \quad g_l \leftarrow \text{JVP}(f, h_l, p_l)$       via forward-mode AD
 8:             $\widehat{J}_l \leftarrow g_l p_l^T$           (rank-1 estimate)
 9:         **end for**
10:         **2) Feedback refinement:**
11:         **for** $l = 1$ to $L$ **do**
12:             compute per-column cosines: $\bar{c}_l = \frac{1}{n_l} \sum_{k=1}^{n_l} \cos(B_l[:, k], \widehat{J}_l[:, k])$
13:             $L_{\text{align}} \leftarrow 1 - \bar{c}_l$
14:             $B_l \leftarrow B_l - \eta_B \nabla_{B_l} L_{\text{align}}$
15:             **for** $k = 1$ to $n_l$ **do**
16:                 $B_l[:, k] \leftarrow B_l[:, k]/\|B_l[:, k]\|$
17:             **end for**
18:         **end for**
19:         **3) DFA-style weight update:**
20:         $\nabla \mathcal{L}_L \leftarrow \partial \mathcal{L}/\partial a_L$
21:         **for all** $l \in [1, L]$ **do**
22:             Parallel projection (via equation 2): $\delta a_l = (B_l \nabla \mathcal{L}_L) \odot \sigma_l'(a_l)$
23:             Weight update: $\delta W_l \leftarrow -\eta \, \delta a_l h_{l-1}^T \quad W_l \leftarrow W_l + \delta W_l$
24:         **end for**
25:     **end for**
26:     **4) Occasional BP calibration:**
27:     **if** epoch mod $T = 0$ **then**
28:         Sample one small minibatch
29:         Perform one full BP pass on it
30:         Update $\{W_l\}$ by standard gradient descent
31:     **end if**
32: **end for**

---

## 4 EMPIRICAL EVALUATION

We evaluate GrAPE in a variety of settings, with shallow and deep image classification models, as well as transformers for language modeling. We empirically show that GrAPE consistently outperforms

Table 1: Performances of a shallow convolutional network (CNN) and a 3 layer Multi Layer Perceptron (MLP) trained on the MNIST and CIFAR10 datasets with different learning algorithms (in percentages).

| Method | Parallelizable | MNIST | | CIFAR10 | | CIFAR100 | |
|---|---|---|---|---|---|---|---|
| | | MLP | CNN | MLP | CNN | MLP | CNN |
| BP | No | $98.73 \pm 0.04$ | $99.03 \pm 0.02$ | $54.09 \pm 0.14$ | $74.66 \pm 0.08$ | $28.18 \pm 0.45$ | $44.22 \pm 0.19$ |
| FA | No | $98.36 \pm 0.04$ | $98.7 \pm 0.07$ | $52.18 \pm 0.15$ | $71.05 \pm 0.18$ | $24.54 \pm 0.22$ | $35 \pm 0.27$ |
| DRTP | Yes | $95.7 \pm 0.12$ | $98.5 \pm 0.17$ | $47.55 \pm 0.12$ | $64.73 \pm 0.62$ | $18.63 \pm 0.43$ | $30.54 \pm 0.12$ |
| DFA | Yes | $98.21 \pm 0.07$ | $98.6 \pm 0.04$ | $51.32 \pm 0.32$ | $69.34 \pm 0.4$ | $22.44 \pm 0.23$ | $34.53 \pm 0.42$ |
| PEPITA | Yes | $98.01 \pm 0.08$ | NA | $52.01 \pm 0.13$ | NA | $21.87 \pm 0.25$ | NA |
| GrAPE (ours) | Yes | $98.53 \pm 0.02$ | $98.8 \pm 0.01$ | $\mathbf{53.4 \pm 0.04}$ | $\mathbf{73.1 \pm 0.23}$ | $\mathbf{26.22 \pm 0.33}$ | $\mathbf{38.0 \pm 0.31}$ |

DFA and other methods, even without BP calibration. Introducing the occasional calibration step improves both DFA and GrAPE, and brings GrAPE much closer to BP.

A crucial requirement for any BP alternative is rigorous implementation and evaluation under the same settings as BP. Since these methods are still nascent, fair and reproducible comparisons are essential. Although FDFA (Bacho & Chu, 2024) also combines random feedback with forward-mode AD, it does not, to our knowledge, provide an optimization-theoretic convergence argument. Our attempts to reproduce their reported numbers across CIFAR-100 settings did not match the paper's figures using the authors' code or a BioTorch reimplementation; details are provided in the Appendix A.4.

## 4.1 EXPERIMENTAL SETTING

We implement our method in Biotorch (Sanfiz & Akrout, 2021) to ensure full transparency and to leverage its existing feedback-alignment and forward-mode hooks. All hyper-parameters (learning rates, schedulers, etc.) are detailed in the Appendix E, and code will be released upon publication. In our current Biotorch implementation, computations are largely serialized on a single GPU and we do not yet exploit true layer-parallel scheduling. We chose Biotorch to ensure a fair comparison with existing baselines and a standardized way of benchmarking our method within the same framework. In this regime, GrAPE incurs a modest 6–20% wall-clock overhead per training step compared to DFA or BP, depending on architecture and dataset, due to the extra forward-mode JVPs and local alignment updates.

To complement backward hooks in convolutions, we use PyTorch's low-level `conv2d_input`, `conv2d_weight`, `conv2d_bias` from `torch.nn.grad` and the JVP routines from FwAD, ensuring correct gradient estimates in convolutional layers without any backward pass. All experiments were conducted on a NVIDIA A100 GPU. We test our method on the following setups:

**Shallow architectures.** We first validate on a 3-layer MLP (hidden size 1024) and a LeNet-5–style CNN across MNIST, CIFAR-10, and CIFAR-100. Baselines follow the strongest BP-free comparators in Srinivasan et al. (2023): FA, DFA, DRTP (Frenkel et al., 2021) and PEPITA (Dellaferrera & Kreiman, 2022) with the recommended variance-reduction tweaks from Srinivasan et al. (2023). Consistent with prior reports, DRTP/PEPITA do not scale to deep CNNs, hence we exclude them from AlexNet/VGG/ResNet tables. The results are reported Table 1.

**Deeper convnets with BP calibration.** Next, we tackle AlexNet and VGG-16 on CIFAR-100 – architectures on which vanilla DFA catastrophically fails (Launay et al., 2019). Here we inject one backpropagation update on a randomly selected mini-batch every $T$ epochs (for both GrAPE and DFA as a "calibrated" control). Surprisingly, this sparse BP step alone recovers a large fraction of DFA's gap with BP and allows GrAPE to attain really close performance when compared to BP. The results are reported in Table 2 and Figure 2.

**Cost of BP calibration.** In all calibrated settings, a BP step consists of a single full backward pass on one mini-batch every $T$ epochs. On CIFAR-100 with ResNet-20 and batch size 256, this corresponds to one calibration batch out of $\approx 195$ per epoch; for $T = 1$ this is about $0.5\%$ of the backward passes used by standard BP, and for larger $T$ the overhead is reduced proportionally. Thus, even at $T = 1$ the calibration cost is negligible compared to the bulk of GrAPE or DFA updates.

**Modern architectures with BP calibration.** Finally, we scale to a Transformer-Base on WikiText-103, following exactly the protocols of Launay et al. (2020). We adopt macro (one feedback per encoder block) and micro (one per sub-layer) feedback approaches, replacing each fixed feedback by a learned one via the same local cosine-alignment, with no change to forward or attention internals;

Table 2: Performances of AlexNet and VGG-16 models trained on CIFAR-100 with different learning algorithms.

| Method | AlexNet | VGG-16 |
|---|---|---|
| BP | $64.61 \pm 0.29$ | $70.33 \pm 0.61$ |
| DFA | $42.59 \pm 0.34$ | $1.00 \pm 0.00$ |
| DFA + calibration ($T = 1$) | $49.37 \pm 0.16$ | $29.40 \pm 0.82$ |
| GrAPE | $45.45 \pm 0.20$ | $32.40 \pm 0.32$ |
| GrAPE + calibration ($T = 1$) | $\mathbf{62.63} \pm 0.52$ | $\mathbf{56.93} \pm 0.11$ |

BP inside attention layers remains as in Launay et al. (2020), depending on the specific setting. We also scale to deep networks such as ResNet-20/56 on CIFAR-100 per canonical practice in He et al. (2016) and also on Tiny ImageNet, using it as a compute-efficient proxy that preserves ImageNet-like statistics, mirroring full-ImageNet trends (Shleifer & Prokop, 2019). Once again on these deep network, we apply a BP calibration step on a single mini batch every $T$ epochs to both GrAPE and DFA. We report averages over 10 independent runs of each method's best checkpoint, with standard deviations to reflect stability. The results are reported in Tables 3 and 4.

**Potential gains under layer parallelism.** To probe the potential gains under actual layer parallelism, we also implemented a small prototype on a Transformer with hidden size 128, depths 2/4/8, batch size 256 and sequence length 64 on a single NVIDIA A100. Using Python-level CUDA streams and a simple double-forward trick to compute JVPs (duplicating the batch and perturbing the duplicate to compute JVPs), the mean time per batch was roughly three times lower for GrAPE versus BP (Table 5 in Appendix D). These numbers are conservative (no kernel fusion or custom kernels), but they illustrate that once layer-parallelism is exploited, GrAPE can reduce wall-clock time relative to sequential BP, especially at larger depths.

## 4.2 Results Analysis

**Shallow architectures (Table 1)** In our preliminary small-scale experiments on a 3-layer MLP (hidden size 1024) and a LeNet-5–style CNN across MNIST, CIFAR-10 and CIFAR-100 (Table 1), vanilla GrAPE surpasses every other method – FA, DFA, DRTP and PEPITA – without any BP calibration. This shows that in low-dimensional or shallow settings, GrAPE's forward-gradient estimates are sufficiently accurate to drive learning effectively without full backpropagation updates.

**AlexNet and VGG-16 (Table 2 and Figure 2)** BP achieves the highest accuracies, with $64.6\% \pm 0.3$ on AlexNet and $70.3\% \pm 0.6$ on VGG-16. Uncalibrated DFA performs poorly (only $42.6\% \pm 0.3$ on AlexNet and $1.0\%$ on VGG-16). Introducing one BP calibration per epoch ($T = 1$) boosts DFA by over 6 points on AlexNet and nearly 30 points on VGG-16. GrAPE without calibration starts higher ($45.5\% \pm 0.2$ on AlexNet, $32.4\% \pm 0.3$ on VGG-16), but with $T = 1$ it almost matches BP, reaching $62.6\% \pm 0.5$ and $56.9\% \pm 0.1$, closely trailing the BP curve in Figure 2.

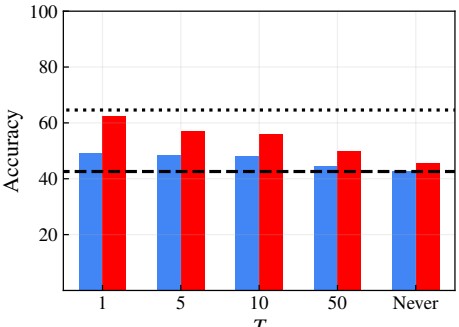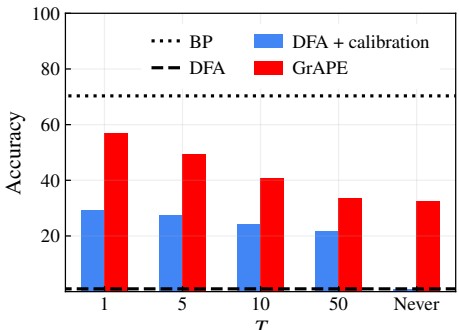

Figure 2: Accuracy vs. calibration interval $T$ for AlexNet (left) and VGG-16 (right) on CIFAR-100. $T$ = number of epochs between two BP calibration steps (i.e., one calibration every $T$ epochs). We compare BP, DFA, DFA + calibration ($T$), GrAPE, and GrAPE + calibration ($T$).

Table 3: Performance (%) on ResNet-20 and ResNet-56, for CIFAR-100 and Tiny ImageNet. Here, $T$ specifies the number of epochs between BP updates on a randomly selected mini batch.

| Method | $T$ | ResNet-20 | | ResNet-56 | |
|---|---|---|---|---|---|
| | | CIFAR-100 | Tiny ImageNet | CIFAR-100 | Tiny ImageNet |
| BP | | $68.72 \pm 0.14$ | $51.66 \pm 0.74$ | $71.42 \pm 0.60$ | $56.86 \pm 0.83$ |
| DFA | | $20.94 \pm 0.19$ | $14.18 \pm 0.11$ | $24.29 \pm 0.41$ | $15.31 \pm 0.05$ |
| GrAPE (ours) | | $24.28 \pm 0.36$ | $18.63 \pm 0.37$ | $29.33 \pm 0.63$ | $20.15 \pm 0.12$ |
| | 1 | $59.80 \pm 0.55$ | $46.13 \pm 0.29$ | $62.43 \pm 0.15$ | $48.39 \pm 0.48$ |
| DFA + calibration | 5 | $55.28 \pm 0.22$ | $43.56 \pm 0.57$ | $61.40 \pm 0.81$ | $47.00 \pm 0.34$ |
| | 10 | $53.79 \pm 0.29$ | $44.21 \pm 0.22$ | $60.29 \pm 0.40$ | $46.92 \pm 0.26$ |
| | 50 | $30.06 \pm 0.93$ | $20.78 \pm 0.71$ | $53.91 \pm 0.54$ | $22.83 \pm 0.78$ |
| | 1 | $\mathbf{64.82} \pm 0.55$ | $\mathbf{48.96} \pm 0.21$ | $\mathbf{66.92} \pm 0.26$ | $\mathbf{51.68} \pm 0.48$ |
| GrAPE + calibration | 5 | $63.09 \pm 0.53$ | $45.17 \pm 0.17$ | $65.92 \pm 0.62$ | $49.02 \pm 0.73$ |
| | 10 | $61.15 \pm 0.21$ | $44.44 \pm 0.28$ | $65.75 \pm 0.59$ | $47.72 \pm 0.21$ |
| | 50 | $36.79 \pm 0.62$ | $22.15 \pm 0.58$ | $56.47 \pm 0.54$ | $24.40 \pm 0.47$ |

Table 4: Best validation perplexity after 20 epochs of a Transformer trained on WikiText-103 (lower is better). $T$ specifies the number of epochs between BP updates on a randomly selected mini batch.

| | BP | DFA | GrAPE | DFA + calibration | | | GrAPE + calibration | | |
|---|---|---|---|---|---|---|---|---|---|
| $T$ | | | | 1 | 5 | 10 | 1 | 5 | 10 |
| Macro | 29.8 | 52.0 | 42.3 | 42.7 | 48.2 | 50.1 | **33.1** | 37.8 | 40.4 |
| Micro | | 93.3 | 81.1 | 78.8 | 87.9 | 90.5 | **67.3** | 73.7 | 78.2 |

**ResNet-20 and ResNet-56 (Table 3)** On CIFAR-100 with ResNet-20, uncalibrated DFA achieves only $20.9\% \pm 0.2$ and GrAPE $24.3\% \pm 0.4$, compared to BP's $68.7\% \pm 0.1$. A single BP calibration every epoch ($T = 1$) elevates DFA to $59.8\% \pm 0.6$ and GrAPE to $64.8\% \pm 0.6$, closing most of the gap with BP. As $T$ increases to 5, 10, and 50 epochs, both methods gradually lose accuracy, highlighting the need for frequent calibration. Similar trends hold for ResNet-56 and on Tiny ImageNet: GrAPE with $T = 1$ consistently outperforms calibrated DFA and approaches BP performance.

**Transformer-Base on WikiText-103 (Table 4)** For the language modeling task, uncalibrated DFA yields perplexities of 52.0 (Macro) and 93.3 (Micro), while GrAPE starts at 42.3 and 81.1. With $T = 1$, DFA improves to 42.7/78.8, but GrAPE reaches 33.1/67.3, cutting its gap with BP (29.8) by nearly half. Calibration intervals of 5 and 10 epochs result in progressively worse perplexities, confirming that frequent BP injections are key to maintaining model quality.

**Critical role of periodic BP calibration** All experiments indicate that regularly performing one BP can greatly improve the training of both DFA and GrAPE. It seems particularly critical for larger and deeper models (e.g. VGG-16, Transformers and ResNets). One potential reason is that the variance of the forward gradient grows linearly with the hidden dimension: larger models lead to more noisy gradient estimates, which makes training more unstable. However, ablation studies with no calibration steps show that the GrAPE learning rule with adaptive feedback matrices is fundamentally better than DFA. In some cases (eg. Transformer, VGG-16), GrAPE with no calibration even outperforms DFA with calibration. With matching calibration frequencies, GrAPE also consistently outperforms DFA.

Across all architectures and tasks, periodic BP calibration on a single random batch effectively bridges most of the performance gap between approximate methods (DFA, GrAPE) and full backpropagation. In particular, GrAPE with $T = 1$ matches BP performance very closely while relying on update rules with a shorter arithmetic critical path than sequential BP; in our current serialized implementation this manifests as a modest 6–20% per-step overhead, but simple layer-parallel prototypes already show potential wall-clock speedups at larger depths (Table 5).

## 5 Discussion and Limitations

Our empirical results demonstrate that GrAPE achieves state-of-the-art performance among feedback-alignment variants in shallow models (Table 1), and recovers the majority of BP's accuracy on deep convnets with at most a single BP calibration per epoch (Tables 2, 3). Furthermore, this BP calibration step yields substantial perplexity gains on Transformers (Table 4). Nonetheless, several important caveats remain. First of all, because our BioTorch-based prototype serializes updates, we do not report realized layer-parallel wall-clock gains: these are deferred to future purpose-built kernels.

**Necessity and cost of BP calibration** While vanilla GrAPE suffices for low-dimensional / shallow settings, deep or wide networks depend critically on periodic BP updates: calibration frequency $T = 1$ consistently outperforms $T > 1$, but even a single-batch BP step might incur nontrivial overhead compared to fully local methods. Designing adaptive or event-driven calibration schedules – triggered by alignment metrics rather than fixed epochs – could reduce this cost, as well as an advanced mechanism to select the samples contained in the considered batch.

This calibration can be interpreted in a federated-learning analogy: each layer's GrAPE update plays the role of independent "local training" on edge devices, using only cheap, parallelizable forward-gradient corrections; the occasional full BP calibration then acts like a central-server aggregation step, collecting the current model state, computing an exact gradient "global update," and redistributing the realigned weights back to all devices. In this way, we retain fully parallel local updates most of the time, yet periodically synchronize via a trusted central pass to ensure that all feedback matrices stay coherently aligned across the entire network.

**Variance in forward-mode estimates** Our Jacobian estimate is unbiased but exhibits a variance proportional to the model dimension (Section 4). In wide layers, the alignment signal may collapse and noise dominates, necessitating frequent calibration steps and reducing the impact of vanilla GrAPE. Future work could explore multiple perturbation directions at each forward or lightweight local losses (as in (Fournier et al., 2023)) to reduce variance, at the expense of additional computation.

**Convergence guarantees** Our theoretical analysis provides estimator-level Jacobian alignment guarantees and recalls a standard conditional convergence result under a positive expected-cosine assumption on the update direction. In deep nonconvex landscapes, this leaves open the risk of saddle-point or poor-quality minima. Incorporating second-order information (e.g. diagonal Hessian approximations via forward-mode AD) may strengthen convergence toward high-quality minima.

**Architectural generality** We validated GrAPE on multiple types of networks without designing adapted shapes for the feedback matrices. Imposing structures on the feedback matrices to respect the inherent composition of the Jacobians (for example block-diagonal for convolutions) could be a promising avenue to reduce reliance on BP calibration and further close the gap with full BP.

In summary, GrAPE moves us closer to truly parallel, local learning methods, but fully matching BP on large-scale, modern architectures will require advances in variance reduction, adaptive calibration, and architecture-aware feedback design.

## 6 Conclusion

We have presented GrAPE, a novel feedback-alignment algorithm that replaces the conventional backward pass with parallelizable feedback projections, aligned with the gradient direction. By computing cheap rank-1 Jacobian approximations during the forward pass and injecting occasional backpropagation updates, GrAPE combines the parallelism of forward-only methods with an accuracy close to that of standard backpropagation. Our empirical evaluation confirms that in shallow settings, GrAPE already outperforms all existing feedback-alignment variants without any BP calibration. More importantly, on deep convolutional and residual networks and on a Transformer-Base model, a backpropagation step per epoch on a single batch is sufficient to recover nearly the same accuracy or perplexity as standard BP, closing most of the gap that purely local methods leave behind.

Although these results bring us significantly closer to truly parallel learning, several avenues remain to fully match BP on large, modern architectures. First, adaptive calibration strategies could reduce the overhead of intermittent BP steps. Second, variance-reduction techniques may further stabilize training in very wide layers. Finally, extending GrAPE's feedback projections to exploit the specific structure of the considered layers could also help to narrow the remaining performance gap without relying on frequent BP resets. By uniting forward-mode gradient estimates with targeted backpropagation corrections, GrAPE lays the groundwork for scalable, parallel and efficient training of deep neural networks.

ACKNOWLEDGMENTS

This work was granted access to the HPC resources of IDRIS under the allocation A0191016927 made by GENCI. This work has received support from the French government, managed by the National Research Agency, under the France 2030 program with the reference "PR[AI]RIE-PSAI" (ANR-23-IACL-0008) and "PEPR-SHARP" (ANR-23-PEIA-0008).

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

# A  DETAILED RELATED WORK

## A.1  LEARNING WITH RANDOM FEEDBACK

**Feedback Alignment (FA)**  introduces a paradigm-shifting and biologically plausible alternative to gradient backpropagation (Lillicrap et al., 2016). At the core of this method, random feedback matrices simplify the weight update process and break the symmetry. FA sequentially multiplies the output error by the random feedback matrices in order to obtain the error at each layer, which is in turn used to update the weights. The random matrices replace the transpose of the weights in the original backpropagation equation. However, the sequential aspect of the update remains, as the error is still propagated from layer to layer through the random feedback projections. With $B_l$ the fixed random feedback matrix of the $l$-th layer, the update can be computed as follows:

$$\delta a_L = (B_L e) \odot \sigma'_L(a_L), \text{ with}$$
$$\delta a_l = (B_l \delta a_{l+1}) \odot \sigma'_l(a_l), \quad \forall l \in [1, L-1].$$

This method draws its inspiration from biological neural networks, which do not exhibit symmetric weight transport during learning (Lillicrap et al., 2020), making this learning paradigm more biologically plausible.

**Direct Feedback Alignment (DFA)**  Nøkland (2016) goes one step further: during training, the error signal is directly projected from the output layer to all hidden layers without modification or intermediate calculations. With this additional simplification, the updates can be easily parallelized:

$$\forall l \in [1, L], \delta a_l = (B_l e) \odot \sigma'_l(a_l). \tag{7}$$

FA and DFA have been shown to perform reasonably well on certain tasks and architectures, especially when considering its profound shift from the backpropagation method. While they do not consistently outperform or even compete with backpropagation, their simplicity along with biological plausibility stimulate research to scale up their use, as well as exploration to understand their key limitations. Bartunov et al. (2018) for example, show empirically that FA variants perform significantly worse on CIFAR-10 and ImageNet than BP, for convolutional networks in particular.

This is further analyzed by Launay et al. (2019) in which they exhibit a bottleneck effect that prevents learning in narrow layers, especially in the case of convolutional networks. As a workaround, some variants of FA showed promising performances on deep CNNs (Moskovitz et al., 2018). A seminal work by Akrout et al. (2019) for instance used weight mirroring to adapt the feedback matrices during training, matching BP performances. However these approaches stay sequential, and similar approaches to DFA with target projection, such as DRTP (Frenkel et al., 2021), do not compete with BP on more complex convolutional networks.

It has also been empirically verified (Launay et al., 2020) that learning under synaptic asymmetry with DFA is possible even with Transformers (Vaswani et al., 2017). In this particular work, the training of the Transformer with DFA is done according to two settings: the 'macro' setting in which the feedback is applied after every encoder block and the 'micro' setting, in which it is done after every layer in those blocks. As explained by Launay et al. (2020) in Appendix D, backpropagation through the attention mechanism itself still happens even in the 'micro' setting, meaning that the training process still relies on BP within transformer layers without reaching the same perplexity levels as full BP training.

In their papers, Nøkland (2016) and Refinetti et al. (2021) analyze the underlying dynamics in the FA-like algorithms to better explain their ability and inability to learn. A key lesson is that the angle between the update and the true gradient must be lower than $\pm \pi/2$. Equivalently, if we denote $\omega_l$ this angle, and $B_l$ the $l$-th layer's feedback matrix, the following inequality must hold:

$$\forall l \in [1, L], \quad \cos(\omega_l) = \frac{\nabla \mathcal{L}_l^T B_l e}{\|\nabla \mathcal{L}_l\| \cdot \|B_l e\|} > 0$$

We recognize a particular case of the Zoutendijk theorem (Nocedal & Wright, 1999), which ensures global convergence when the search direction makes an angle with the steepest descent direction bounded away from $\pi/2$. This theorem requires that the step length satisfies either the Goldstein or

strong Wolfe conditions, and this is typically the case with standard learning rates. However, let us stress that the considered convergence is towards local minima and stationary points.

As previously mentioned, the recent work of (Akrout et al., 2019) revisits the idea to learn the feedback by emulating a Kolen-Pollack algorithm (Kolen & Pollack, 1994) or with an estimate of the transpose matrix. This idea facilitates the learning process of FA while reducing the angles $\omega_l$. This first attempt clearly shows that adaptive feedback matrices enable the learning of networks on which FA previously failed. It also emphasizes the importance of Zoutendijk's theorem, even though the sequential learning process inherited from the FA still inhibits the potential improvements.

## A.2   Forward only calculations

A promising avenue toward *backward-free* training is the *double-forward* approach, in which two forward passes are used: the *first* forward pass updates an auxiliary or feedback mechanism, while the *second* forward pass computes the weight updates. The recent paper (Srinivasan et al., 2023) follows this trend and exhibits similarities between two forward-only frameworks, Forward-Forward and PEPITA (Hinton, 2022; Dellaferrera & Kreiman, 2022). They also show that such algorithms can be approximated by a form of feedback alignment with adaptive feedback (AF) weights, modulated by the upstream network weights. The PEPITA learning rule essentially is: $\delta W_l = (h_l - h_l^{err}) \odot (h_{l-1}^{err\,T})$, with $h_0^{err} = x - Fe$, where $F$ can be viewed as a feedback mapping on the input. Srinivasan et al. (2023) showed that PEPITA implements a feedback-alignment learning algorithm with an adaptive feedback matrix that depends on the forward weights when $F$ is computed with weight mirroring. Although promising, this method fails to scale to networks deeper than 5 layers.

**Forward Gradient (FG)**, introduced by Silver et al. (2021) and Baydin et al. (2022) employs Forward-Mode Automatic Differentiation (FwAD) as proposed in Margossian (2019) to estimate gradients solely through forward passes. Focusing on these forward calculations, recent works explore unbiased estimations of the gradients, thanks to directional derivatives (Fournier et al., 2023; Baydin et al., 2022). These gradients are then used to update the weights like in standard BP, without needing an explicit backward pass. The essential idea in FG descent is that given a direction vector $\mathbf{u} \in \mathbb{R}^m$, computing the Jacobian-vector product (JVP) of the gradient of the loss along $\mathbf{u}$ gives the gradient of the loss function according the direction given by $\mathbf{u}$. This is defined as:

$$\nabla \mathcal{L} \, \mathbf{u} \equiv \lim_{\delta \to 0} \frac{\mathcal{L}(\theta + \delta \, \mathbf{u}) - \mathcal{L}(\theta)}{\delta},$$

at the parameter point $\theta$. This is used to estimate partial derivatives of the loss with respect to subsets of parameters or activations, along random directions.

Although Baydin et al. (2022) showed that sampling the perturbations $\mathbf{u}$ in the weight space can provide unbiased gradient estimates, Ren et al. (2022) revealed poor scalability when the number of parameters is large. To address this, they proposed to draw perturbations in the activation space, inspired by Le Cun et al. (1988) and Widrow & Lehr (1990). Since the total number of neurons $n$ is usually much smaller than the total count of parameters, sampling $\mathbf{u}_l \in \mathbb{R}^{n_l}$ for each layer $l$ can substantially reduce both the cost and variance of gradient estimation.

However, recent advances (Fournier et al., 2023) show that in modern settings, even variance-reducing techniques with local losses do not allow to reach performance on-par with standard backpropagation.

## A.3   Reminders on Zoutendijk's theorem

Let $f : \mathbb{R}^n \to \mathbb{R}$ be a twice continuously differentiable function, bounded below on $\mathbb{R}^n$. Consider the iteration

$$x_{k+1} = x_k + \alpha_k p_k,$$

where each $p_k$ is a descent direction (i.e. $\nabla f(x_k)^\top p_k < 0$) and the step length $\alpha_k > 0$ satisfies the Wolfe conditions:

$$f(x_k + \alpha_k p_k) \leq f(x_k) + c_1 \, \alpha_k \, \nabla f(x_k)^\top p_k, \quad 0 < c_1 < c_2 < 1, \tag{8}$$

$$\nabla f(x_k + \alpha_k p_k)^\top p_k \geq c_2 \, \nabla f(x_k)^\top p_k. \tag{9}$$

**Theorem A.1** (Zoutendijk). *Under these assumptions, the series*

$$\sum_{k=0}^{\infty} \cos^2\theta_k \left\| \nabla f(x_k) \right\|^2, \quad where \quad \cos\theta_k = \frac{-\nabla f(x_k)^\top p_k}{\left\| \nabla f(x_k) \right\| \left\| p_k \right\|},$$

*converges. In particular,*

$$\liminf_{k\to\infty} \|\nabla f(x_k)\| = 0.$$

*Sketch of Proof.* Starting from the decrease guaranteed by the Armijo condition equation 8 and invoking the curvature condition equation 9, one shows (see Nocedal & Wright (1999), Chapter 3) that there exists a constant $M > 0$ such that

$$\alpha_k\left(-\nabla f(x_k)^\top p_k\right) \;\geq\; M \cos^2\theta_k \left\| \nabla f(x_k) \right\|^2.$$

Summing over $k$ then yields the claimed convergence of the series. $\square$

Zoutendijk's theorem thus states that if each search direction $p_k$ remains positively aligned with the negative gradient, i.e. $\nabla f(x_k)^\top p_k < 0$, and the step lengths $\alpha_k > 0$ satisfy the Wolfe conditions equation 8–equation 9, then $\sum_{k=0}^{\infty} \cos^2\theta_k \|\nabla f(x_k)\|^2 < \infty$, which in turn implies

$$\liminf_{k\to\infty} \|\nabla f(x_k)\| = 0.$$

### A.4 FDFA REPRODUCTION DETAILS

We deliberately excluded Bacho & Chu (2024) because our attempts to reproduce their results revealed multiple inconsistencies:

- On AlexNet (CIFAR-100), their code yields $55.03\% \pm 0.16$ accuracy over 10 runs, instead of the $57.27\% \pm 0.11$ they report.
- Their DFA baseline (from the Webster et al. (2021) code) achieves $48.03\% \pm 0.61$, yet they report only $35.75\% \pm 0.58$. This discrepancy appears to arise from applying batch–norm exclusively to FDFA.
- Re-implementing their method in BioTorch with their hyperparameters gives no improvement over standard DFA ($33.59\%$ for FDFA vs. $42.59\%$ for DFA).
- Finally, integrating GrAPE into their code gives $60.35\% \pm 0.26$ on AlexNet (no augmentation), essentially matching BP, which conflicts with the known gap between BP and DFA-style methods.

Collectively, these points suggest unintended reliance on backpropagation in their pipeline and cherry-picked reporting, undermining reproducibility and justifying our decision to omit FDFA from our baselines.

**Novelty of GrAPE.** While previous works combine DFA with forward-mode AD, GrAPE is the first to:

- derive feedback updates from a *cosine-similarity alignment loss* grounded in the intuition of Zoutendijk's theorem and supported by estimator-level Jacobian alignment guarantees,
- demonstrate scalability to deep architectures (VGG, ResNet, Transformers) with rigorous empirical validation,
- provide estimator-level Jacobian alignment guarantees together with extensive benchmarks using verified implementations (e.g. BioTorch).

While we are open to citing FDFA in future versions for completeness, we believe that its omission in the present version is fully justified and does not diminish the novelty or relevance of our contribution.

## B FORWARD-GRADIENT ESTIMATOR: FROBENIUS COSINE AND CONCENTRATION

In this appendix we justify the lower bound on the expected Frobenius cosine between the true Jacobian $\mathcal{J}_l$ and its rank-1 estimator $\widehat{\mathcal{J}}_l = (\mathcal{J}_l \mathbf{p})\mathbf{p}^\top$, where $\mathbf{p} \sim \mathcal{N}(0, I_{n_l})$, together with an $O(1/\sqrt{B})$ concentration rate for the batched estimator.

## B.1 REPRESENTATION VIA THE SPHERE

Let $\mathbf{p} \sim \mathcal{N}(0, I_{n_l})$. We write

$$\mathbf{p} = r\,\mathbf{s}, \qquad r := \|\mathbf{p}\|, \quad \mathbf{s} := \mathbf{p}/\|\mathbf{p}\|.$$

It is classical that $\mathbf{s}$ is uniform on the unit sphere $S^{n_l-1}$ and independent of $r$. For the rank-1 estimator $\widehat{\mathcal{J}_l} = (\mathcal{J}_l \mathbf{p})\mathbf{p}^\top$, a straightforward computation shows that the Frobenius cosine satisfies

$$\cos_F\big(\mathcal{J}_l, \widehat{\mathcal{J}_l}\big) = \frac{\langle \mathcal{J}_l, \widehat{\mathcal{J}_l}\rangle_F}{\|\mathcal{J}_l\|_F \|\widehat{\mathcal{J}_l}\|_F} = \frac{\|\mathcal{J}_l \mathbf{p}\|}{\|\mathcal{J}_l\|_F \|\mathbf{p}\|} = \frac{\|\mathcal{J}_l \mathbf{s}\|}{\|\mathcal{J}_l\|_F},$$

so that

$$\mathbb{E}\Big[\cos_F\big(\mathcal{J}_l, \widehat{\mathcal{J}_l}\big)\Big] = \frac{1}{\|\mathcal{J}_l\|_F} \mathbb{E}_{\mathbf{s}}\|\mathcal{J}_l \mathbf{s}\|, \qquad \mathbf{s} \sim \mathrm{Unif}(S^{n_l-1}). \tag{10}$$

## B.2 LOWER BOUND VIA THE TOP SINGULAR VECTOR

Let the singular value decomposition of $\mathcal{J}_l$ be $\mathcal{J}_l = U\Sigma V^\top$, and denote by $\sigma_{\max} = \|\mathcal{J}_l\|_2$ the largest singular value, with associated right singular vector $\mathbf{v}_1$ and left singular vector $\mathbf{u}_1$. For any unit vector $\mathbf{u}$ and any $x$, $\|x\| \geq |\mathbf{u}^\top x|$; choosing $\mathbf{u} = \mathbf{u}_1$ yields

$$\|\mathcal{J}_l \mathbf{s}\| \geq \big|\mathbf{u}_1^\top \mathcal{J}_l \mathbf{s}\big| = \big|\sigma_{\max} \mathbf{v}_1^\top \mathbf{s}\big| = \|\mathcal{J}_l\|_2 \big|\mathbf{v}_1^\top \mathbf{s}\big|.$$

Taking expectations over $\mathbf{s}$ gives

$$\mathbb{E}_{\mathbf{s}}\|\mathcal{J}_l \mathbf{s}\| \geq \|\mathcal{J}_l\|_2 \mathbb{E}_{\mathbf{s}}\big|\mathbf{v}_1^\top \mathbf{s}\big|.$$

By rotational invariance of the uniform distribution on the sphere, the scalar $\mathbf{v}_1^\top \mathbf{s}$ has the same distribution as the first coordinate $s_1$ of $\mathbf{s} \sim \mathrm{Unif}(S^{n_l-1})$. Hence

$$\mathbb{E}_{\mathbf{s}}\|\mathcal{J}_l \mathbf{s}\| \geq \|\mathcal{J}_l\|_2 \mathbb{E}|s_1|.$$

A standard computation (properties of the spherical distribution) shows that

$$\mathbb{E}|s_1| = \frac{\Gamma\big(\frac{n_l}{2}\big)}{\sqrt{\pi}\,\Gamma\big(\frac{n_l+1}{2}\big)}.$$

Using classical bounds on ratios of Gamma functions (Gautschi's inequality) one obtains, for all $n_l \geq 2$,

$$\mathbb{E}|s_1| \geq \sqrt{\frac{2}{\pi n_l}}. \tag{11}$$

Combining this with the previous inequality yields

$$\mathbb{E}_{\mathbf{s}}\|\mathcal{J}_l \mathbf{s}\| \geq \sqrt{\frac{2}{\pi n_l}}\,\|\mathcal{J}_l\|_2. \tag{12}$$

Substituting equation 12 into equation 10 yields the lower bound used in the main text:

$$\mathbb{E}\Big[\cos_F\big(\mathcal{J}_l, \widehat{\mathcal{J}_l}\big)\Big] = \frac{1}{\|\mathcal{J}_l\|_F} \mathbb{E}_{\mathbf{s}}\|\mathcal{J}_l \mathbf{s}\| \geq \sqrt{\frac{2}{\pi n_l}}\,\frac{\|\mathcal{J}_l\|_2}{\|\mathcal{J}_l\|_F}, \tag{13}$$

which is strictly positive whenever $\mathcal{J}_l \neq 0$.

## B.3 BATCHED ESTIMATOR AND CONCENTRATION

Consider the batched estimator obtained by averaging $B$ independent rank-1 estimates based on $\mathbf{p}_1, \ldots, \mathbf{p}_B \sim \mathcal{N}(0, I_{n_l})$:

$$\widehat{\mathcal{J}_l}^{(B)} := \frac{1}{B} \sum_{i=1}^{B} (\mathcal{J}_l \mathbf{p}_i)\mathbf{p}_i^\top, \qquad C_i := \cos_F\big(\mathcal{J}_l, \widehat{\mathcal{J}_l}(\mathbf{p}_i)\big).$$

We are interested in the empirical mean

$$\overline{C}_B := \frac{1}{B} \sum_{i=1}^{B} C_i.$$

The map $\mathbf{p} \mapsto C(\mathbf{p}) := \cos_F(\mathcal{J}_l, \widehat{\mathcal{J}}_l(\mathbf{p}))$ is a smooth, bounded function of $\mathbf{p}$ and is Lipschitz with respect to $\mathbf{p}$ with a constant depending only on $\mathcal{J}_l$. By standard concentration results for Lipschitz functionals of Gaussian vectors, there exists a constant $c > 0$ (depending on the dimension and condition numbers of $\mathcal{J}_l$) such that for all $t > 0$,

$$\mathbb{P}\big(|\overline{C}_B - \mathbb{E}C(\mathbf{p})| \ge t\big) \le 2\exp\big(-c\,B\,t^2\big).$$

Equivalently,

$$\mathrm{Std}(\overline{C}_B) = O\Big(\frac{1}{\sqrt{B}}\Big),$$

so the empirical Frobenius cosine concentrates around its expectation at rate $O(1/\sqrt{B})$.

## B.4 CONVERGENCE UNDER POSITIVE EXPECTED COSINE

We recall a standard stochastic-approximation result tailored to our setting.

**Theorem B.1** (Convergence under positive expected cosine). *Let $L : \mathbb{R}^d \to \mathbb{R}$ be differentiable, bounded below, with $L_g$-Lipschitz gradient. Consider*

$$\theta_{t+1} = \theta_t - \eta_t d_t, \qquad g_t := \nabla L(\theta_t),$$

*where $(\mathcal{F}_t)$ is the natural filtration and the step sizes satisfy*

$$\eta_t > 0, \quad \sum_{t=0}^{\infty} \eta_t = \infty, \quad \sum_{t=0}^{\infty} \eta_t^2 < \infty.$$

*Assume there exist constants $\kappa > 0$, $C < \infty$, $\sigma^2 < \infty$ such that, for all $t$,*

$$\mathbb{E}\big[\langle g_t, d_t\rangle \mid \mathcal{F}_t\big] \ge \kappa \|g_t\|^2, \tag{14}$$

$$\mathbb{E}\big[\|d_t\|^2 \mid \mathcal{F}_t\big] \le C\|g_t\|^2 + \sigma^2. \tag{15}$$

*Then*

$$\sum_{t=0}^{\infty} \eta_t\, \mathbb{E}\|g_t\|^2 < \infty \quad \text{and} \quad \liminf_{t\to\infty} \mathbb{E}\|g_t\| = 0.$$

***In particular, the iterates converge to stationarity in expectation.***

*Proof.* Since $L$ has $L_g$-Lipschitz gradient, the standard descent lemma gives, for all $\theta, \theta' \in \mathbb{R}^d$,

$$L(\theta') \le L(\theta) + \langle \nabla L(\theta), \theta' - \theta\rangle + \frac{L_g}{2}\|\theta' - \theta\|^2.$$

Apply this with $\theta = \theta_t$ and $\theta' = \theta_{t+1} = \theta_t - \eta_t d_t$:

$$L(\theta_{t+1}) \le L(\theta_t) + \langle \nabla L(\theta_t), -\eta_t d_t\rangle + \frac{L_g}{2}\eta_t^2\|d_t\|^2$$

$$= L(\theta_t) - \eta_t\langle g_t, d_t\rangle + \frac{L_g}{2}\eta_t^2\|d_t\|^2.$$

Taking conditional expectation w.r.t. the filtration $\mathcal{F}_t$,

$$\mathbb{E}\big[L(\theta_{t+1}) \mid \mathcal{F}_t\big] \le L(\theta_t) - \eta_t\,\mathbb{E}\big[\langle g_t, d_t\rangle \mid \mathcal{F}_t\big] + \frac{L_g}{2}\eta_t^2\,\mathbb{E}\big[\|d_t\|^2 \mid \mathcal{F}_t\big].$$

Using the two assumptions

$$\mathbb{E}\big[\langle g_t, d_t\rangle \mid \mathcal{F}_t\big] \ge \kappa \|g_t\|^2, \qquad \mathbb{E}\big[\|d_t\|^2 \mid \mathcal{F}_t\big] \le C\|g_t\|^2 + \sigma^2,$$

we obtain

$$\mathbb{E}\big[L(\theta_{t+1}) \mid \mathcal{F}_t\big] \leq L(\theta_t) - \eta_t \kappa \|g_t\|^2 + \frac{L_g}{2} \eta_t^2 \left(C\|g_t\|^2 + \sigma^2\right)$$

$$= L(\theta_t) - \left(\kappa - \tfrac{L_g C}{2}\eta_t\right) \eta_t \|g_t\|^2 + \frac{L_g \sigma^2}{2} \eta_t^2.$$

Because $\sum_t \eta_t^2 < \infty$, we have $\eta_t \to 0$ as $t \to \infty$. Hence there exists an index $t_0$ such that for all $t \geq t_0$,

$$\kappa - \frac{L_g C}{2}\eta_t \; \geq \; \frac{\kappa}{2}.$$

For $t \geq t_0$, this yields

$$\mathbb{E}\big[L(\theta_{t+1}) \mid \mathcal{F}_t\big] \; \leq \; L(\theta_t) - \frac{\kappa}{2}\,\eta_t \|g_t\|^2 + \frac{L_g \sigma^2}{2}\,\eta_t^2. \tag{16}$$

Now take the full expectation of equation 16 and use the tower property $\mathbb{E}\,\mathbb{E}[\,\cdot\mid\mathcal{F}_t] = \mathbb{E}[\,\cdot\,]$:

$$\mathbb{E}L(\theta_{t+1}) \; \leq \; \mathbb{E}L(\theta_t) - \frac{\kappa}{2}\,\eta_t\,\mathbb{E}\|g_t\|^2 + \frac{L_g \sigma^2}{2}\,\eta_t^2, \qquad t \geq t_0.$$

Rearranging,

$$\frac{\kappa}{2}\,\eta_t\,\mathbb{E}\|g_t\|^2 \; \leq \; \mathbb{E}L(\theta_t) - \mathbb{E}L(\theta_{t+1}) + \frac{L_g \sigma^2}{2}\,\eta_t^2.$$

Sum this inequality from $t = t_0$ to $T$:

$$\frac{\kappa}{2} \sum_{t=t_0}^{T} \eta_t\,\mathbb{E}\|g_t\|^2 \leq \sum_{t=t_0}^{T} \big(\mathbb{E}L(\theta_t) - \mathbb{E}L(\theta_{t+1})\big) + \frac{L_g \sigma^2}{2} \sum_{t=t_0}^{T} \eta_t^2$$

$$= \mathbb{E}L(\theta_{t_0}) - \mathbb{E}L(\theta_{T+1}) + \frac{L_g \sigma^2}{2} \sum_{t=t_0}^{T} \eta_t^2.$$

By assumption, $L$ is bounded below, say $L(\theta) \geq L_\star$ for all $\theta$. Therefore $\mathbb{E}L(\theta_{T+1}) \geq L_\star$, and we obtain

$$\frac{\kappa}{2} \sum_{t=t_0}^{T} \eta_t\,\mathbb{E}\|g_t\|^2 \; \leq \; \mathbb{E}L(\theta_{t_0}) - L_\star + \frac{L_g \sigma^2}{2} \sum_{t=t_0}^{T} \eta_t^2.$$

Letting $T \to \infty$ and using $\sum_t \eta_t^2 < \infty$ yields

$$\sum_{t=t_0}^{\infty} \eta_t\,\mathbb{E}\|g_t\|^2 < \infty.$$

Adding the finite partial sum over $t < t_0$ shows that

$$\sum_{t=0}^{\infty} \eta_t\,\mathbb{E}\|g_t\|^2 < \infty.$$

It remains to show that this implies $\liminf_{t\to\infty} \mathbb{E}\|g_t\| = 0$. Suppose, for contradiction, that there exists $\varepsilon > 0$ and $t_1$ such that $\mathbb{E}\|g_t\| \geq \varepsilon$ for all $t \geq t_1$. Then $\mathbb{E}\|g_t\|^2 \geq \varepsilon^2$ for all $t \geq t_1$, and therefore

$$\sum_{t=t_1}^{\infty} \eta_t\,\mathbb{E}\|g_t\|^2 \; \geq \; \varepsilon^2 \sum_{t=t_1}^{\infty} \eta_t.$$

By assumption, $\sum_{t=0}^{\infty} \eta_t = \infty$, so the right-hand side diverges, contradicting the finiteness of $\sum_t \eta_t \mathbb{E}\|g_t\|^2$. Hence we must have $\liminf_{t\to\infty} \mathbb{E}\|g_t\| = 0$.

This establishes the claimed convergence to stationarity in expectation. $\qquad\square$

## B.5 A COMPOSITION BOUND FOR FROBENIUS COSINES

We quantify how noise in the JVP estimator and imperfect learning of $B_\ell$ interact.

**Lemma B.2** (Frobenius cosine composition). *Let $B_\ell, \hat{J}_\ell, J_\ell \in \mathbb{R}^{m \times n}$ be nonzero and define*

$$\cos_F(A, B) := \frac{\langle A, B \rangle_F}{\|A\|_F \|B\|_F}, \qquad \langle A, B \rangle_F := \mathrm{Tr}(A^\top B).$$

*Then*

$$\cos_F(B_\ell, J_\ell) \geq \cos_F(B_\ell, \hat{J}_\ell) \cos_F(\hat{J}_\ell, J_\ell) - \sqrt{1 - \cos_F^2(B_\ell, \hat{J}_\ell)} \sqrt{1 - \cos_F^2(\hat{J}_\ell, J_\ell)}.$$

*Consequently, if $\cos_F(B_\ell, \hat{J}_\ell) \geq \alpha_0$ and $\cos_F(\hat{J}_\ell, J_\ell) \geq \beta_0$, then*

$$\cos_F(B_\ell, J_\ell) \geq \gamma_0(\alpha_0, \beta_0) := \alpha_0 \beta_0 - \sqrt{1 - \alpha_0^2} \sqrt{1 - \beta_0^2}.$$

*Proof.* Let $\mathcal{H}$ be a real Hilbert space with inner product $\langle \cdot, \cdot \rangle$ and induced norm $\|\cdot\|$. In our application, $\mathcal{H}$ is the space of matrices with the Frobenius inner product $\langle A, B \rangle_F = \mathrm{Tr}(A^\top B)$, but the argument holds for any Hilbert space.

Let $u, w, v \in \mathcal{H}$ be unit vectors:
$$\|u\| = \|w\| = \|v\| = 1.$$

Define
$$\alpha := \langle u, w \rangle, \qquad \beta := \langle w, v \rangle, \qquad c := \langle u, v \rangle.$$
By Cauchy–Schwarz, $|\alpha|, |\beta|, |c| \leq 1$.

Consider the $3 \times 3$ Gram matrix of $(u, w, v)$:
$$G := \begin{pmatrix} \langle u, u \rangle & \langle u, w \rangle & \langle u, v \rangle \\ \langle w, u \rangle & \langle w, w \rangle & \langle w, v \rangle \\ \langle v, u \rangle & \langle v, w \rangle & \langle v, v \rangle \end{pmatrix} = \begin{pmatrix} 1 & \alpha & c \\ \alpha & 1 & \beta \\ c & \beta & 1 \end{pmatrix}.$$

Since $G$ is a Gram matrix, it is positive semidefinite (PSD), hence $\det(G) \geq 0$.

We compute the determinant explicitly. A direct calculation yields
$$\det(G) = \begin{vmatrix} 1 & \alpha & c \\ \alpha & 1 & \beta \\ c & \beta & 1 \end{vmatrix}$$
$$= -\alpha^2 + 2\alpha\beta c - \beta^2 - c^2 + 1$$
$$= -\left(c^2 - 2\alpha\beta c + (\alpha^2 + \beta^2 - 1)\right).$$

The constraint $\det(G) \geq 0$ is therefore equivalent to
$$c^2 - 2\alpha\beta c + (\alpha^2 + \beta^2 - 1) \leq 0.$$

We view the left-hand side as a quadratic polynomial in $c$,
$$q(c) := c^2 - 2\alpha\beta c + (\alpha^2 + \beta^2 - 1).$$

Since the coefficient of $c^2$ is $1 > 0$, the inequality $q(c) \leq 0$ means that $c$ lies between the two (real) roots of $q$. Compute the discriminant:
$$\Delta = (-2\alpha\beta)^2 - 4(\alpha^2 + \beta^2 - 1)$$
$$= 4\alpha^2\beta^2 - 4(\alpha^2 + \beta^2 - 1)$$
$$= 4\left(\alpha^2\beta^2 - \alpha^2 - \beta^2 + 1\right)$$
$$= 4\left(1 - \alpha^2 - \beta^2 + \alpha^2\beta^2\right)$$
$$= 4(1 - \alpha^2)(1 - \beta^2).$$

Since $|\alpha|, |\beta| \leq 1$, we have $1 - \alpha^2 \geq 0$ and $1 - \beta^2 \geq 0$, so $\Delta \geq 0$ as expected. The roots of $q$ are
$$c_\pm = \alpha\beta \pm \sqrt{(1 - \alpha^2)(1 - \beta^2)}.$$

The inequality $q(c) \leq 0$ therefore implies

$$c_- \leq c \leq c_+,$$

i.e.

$$\alpha\beta - \sqrt{(1-\alpha^2)(1-\beta^2)} \leq c \leq \alpha\beta + \sqrt{(1-\alpha^2)(1-\beta^2)}.$$

In particular, we obtain the desired lower bound

$$c \geq \alpha\beta - \sqrt{1-\alpha^2}\,\sqrt{1-\beta^2}.$$

We now instantiate this with Frobenius-normalized matrices. Let $B_\ell, \hat{J}_\ell, J_\ell \in \mathbb{R}^{m\times n}$ be nonzero, and define

$$u := \frac{B_\ell}{\|B_\ell\|_F}, \qquad w := \frac{\hat{J}_\ell}{\|\hat{J}_\ell\|_F}, \qquad v := \frac{J_\ell}{\|J_\ell\|_F},$$

viewed as elements of the Frobenius inner-product space. Then

$$\alpha = \langle u, w\rangle_F = \cos_F(B_\ell, \hat{J}_\ell), \quad \beta = \langle w, v\rangle_F = \cos_F(\hat{J}_\ell, J_\ell), \quad c = \langle u, v\rangle_F = \cos_F(B_\ell, J_\ell),$$

and the bound above reads

$$\cos_F(B_\ell, J_\ell) \geq \cos_F(B_\ell, \hat{J}_\ell)\cos_F(\hat{J}_\ell, J_\ell) - \sqrt{1 - \cos_F^2(B_\ell, \hat{J}_\ell)}\,\sqrt{1 - \cos_F^2(\hat{J}_\ell, J_\ell)}.$$

This is exactly the claimed inequality. The "$\gamma_0(\alpha_0, \beta_0)$" form in the lemma statement follows by substituting lower bounds $\alpha_0, \beta_0$ for the two intermediate cosines. $\qquad\square$

## C  FEEDBACK MATRICES UPDATE DETAILS

In the main paper, the theoretical alignment measure between a feedback matrix $B_l$ and the JVP-based Jacobian estimator $\widehat{\mathcal{J}}_l$ is the Frobenius cosine

$$\cos_F(B_l, \widehat{\mathcal{J}}_l) = \frac{\langle B_l, \widehat{\mathcal{J}}_l\rangle_F}{\|B_l\|_F \|\widehat{\mathcal{J}}_l\|_F}, \qquad \langle A, B\rangle_F := \mathrm{Tr}(A^\top B).$$

Writing $B_l = [\mathbf{b}_l^1, \ldots, \mathbf{b}_l^{n_l}]$ and $\widehat{\mathcal{J}}_l = [\mathbf{j}_l^1, \ldots, \mathbf{j}_l^{n_l}]$, we can express this as

$$\cos_F(B_l, \widehat{\mathcal{J}}_l) = \frac{\sum_{k=1}^{n_l} \mathbf{b}_l^{k\top}\mathbf{j}_l^k}{\sqrt{\sum_{k=1}^{n_l}\|\mathbf{b}_l^k\|^2}\sqrt{\sum_{k=1}^{n_l}\|\mathbf{j}_l^k\|^2}} = \sum_{k=1}^{n_l} w_k \cos(\mathbf{b}_l^k, \mathbf{j}_l^k),$$

where

$$\cos(\mathbf{b}_l^k, \mathbf{j}_l^k) = \frac{\mathbf{b}_l^{k\top}\mathbf{j}_l^k}{\|\mathbf{b}_l^k\|\,\|\mathbf{j}_l^k\|} \quad \text{and} \quad w_k = \frac{\|\mathbf{b}_l^k\|\,\|\mathbf{j}_l^k\|}{\|B_l\|_F\,\|\widehat{\mathcal{J}}_l\|_F}, \qquad \sum_{k=1}^{n_l} w_k = 1.$$

Thus the Frobenius cosine is a *weighted* average of the columnwise cosines, with weights proportional to the product of column norms. In our implementation we normalize the columns of $B_l$ after each alignment step, and $\widehat{\mathcal{J}}_l$ has columns of the form $\mathbf{j}_l^k = p_k\,g_l$ (with $p_k$ a scalar component of the Gaussian perturbation and $g_l$ a common JVP), so the $\|\mathbf{j}_l^k\|$ differ mainly through $|p_k|$. Since these scalars are i.i.d. and concentrate around their mean, the weights $w_k$ do not vary dramatically across $k$, and the *unweighted* average of per-column cosines provides a simple and effective proxy for $\cos_F(B_l, \widehat{\mathcal{J}}_l)$.

In practice, we therefore minimize the local cosine alignment loss $\mathcal{L}_{\text{align}}(B_l) = 1 - \bar{c}_l$, where

$$\bar{c}_l = \frac{1}{n_l}\sum_{k=1}^{n_l}\cos(\mathbf{b}_l^k, \mathbf{j}_l^k)$$

is the empirical mean of per-column cosines. This loss coincides with the Frobenius cosine up to the weighting discussed above, and is cheaper to compute while still encouraging layerwise alignment.

Because $\mathcal{L}_{\text{align}}$ decomposes as a sum over columns, the gradient can be written column by column. Specifically, for each $k$ we minimize the single-term contribution $\left[1 - \cos(\mathbf{b}_l^k, \mathbf{j}_l^k)\right]$, scaled by $1/n_l$. Let $\eta_{B_l} > 0$ be the learning rate for layer $l$. Then the per-column update can be written as

$$\mathbf{b}_l^k \; \leftarrow \; \mathbf{b}_l^k \; - \; \frac{\eta_{B_l}}{n_l} \, \nabla_{\mathbf{b}_l^k} \Big[1 - \cos\big(\mathbf{b}_l^k, \mathbf{j}_l^k\big)\Big], \quad k = 1, \ldots, n_l.$$

We recall that $\nabla_{\mathbf{x}} \cos(\mathbf{x}, \mathbf{y}) = \frac{\mathbf{y}}{\|\mathbf{x}\|\|\mathbf{y}\|} - \frac{(\mathbf{x}^\top \mathbf{y})\,\mathbf{x}}{\|\mathbf{x}\|^3\,\|\mathbf{y}\|}$, so the gradient of $[1 - \cos(\mathbf{x}, \mathbf{y})]$ is its negative. Each column $\mathbf{b}_l^k$ is thus updated as:

$$
\begin{aligned}
\mathbf{b}_l^k \; &\leftarrow \; \mathbf{b}_l^k \; - \; \frac{\eta_{B_l}}{n_l} \Big[ -\nabla_{\mathbf{b}_l^k} \cos(\mathbf{b}_l^k, \mathbf{j}_l^k) \Big] \\[2mm]
&= \; \mathbf{b}_l^k \; + \; \frac{\eta_{B_l}}{n_l} \left[ \frac{\mathbf{j}_l^k}{\|\mathbf{b}_l^k\| \, \|\mathbf{j}_l^k\|} - \frac{\big(\mathbf{b}_l^{k\top} \mathbf{j}_l^k\big)\, \mathbf{b}_l^k}{\|\mathbf{b}_l^k\|^3 \, \|\mathbf{j}_l^k\|} \right].
\end{aligned}
\tag{17}
$$

Applying this update for each $k = 1, \ldots, n_l$ increases the mean columnwise cosine and thus aligns $B_l$ with $\widehat{\mathcal{J}}_l$. Because $\mathcal{L}_{\text{align}}$ decomposes as a sum of per-column terms, updating all columns in parallel is equivalent to taking a gradient step on $\mathcal{L}_{\text{align}}$ as a whole. We then renormalize each column $\mathbf{b}_l^k$ after this step to keep norms bounded and maintain the interpretation of the cosine as a purely directional alignment measure.

# D  FLOPs, Critical-Path and Time Analysis

## D.1  FLOPs and Critical-Path Analysis

Throughout this subsection we adopt the standard *GEMM view*, modeling each layer's compute as a (batched) matrix-matrix multiply after the usual reshapes/lowering, which yields simple, comparable FLOP counts.

**Setup and notation.** Let the network have layers $\ell = 1, \ldots, L$. Denote by

$$F \; \triangleq \; \sum_{\ell=1}^{L} O\big(n_{\ell-1}\, n_\ell\big)$$

the FLOPs of one *mini-batch* forward pass under the GEMM view (for conv layers this corresponds to the lowered GEMM). Let $d_{\text{out}}$ be the output dimensionality, and define the per-layer costs:

$$C_\ell^{\text{proj}} = O\big(n_\ell\, d_{\text{out}}\big) \quad \text{(feedback projection)}, \qquad C_\ell^{\text{align}} = O\big(n_\ell\, d_{\text{out}}\big) \quad \text{(local alignment update on } B_\ell\text{)}.$$

We model a single Jacobian–vector product (JVP) as an *overhead* of $\alpha$ times the layer's forward cost; writing a layer-averaged $\bar{\alpha}$ gives a total forward+JVP factor $(1 + \bar{\alpha})$.

**Backpropagation (BP).**

$$F_{\text{BP}}^{\text{batch}} \; = \; \sum_{\ell=1}^{L} \Big( O\big(n_{\ell-1} n_\ell\big) + O\big(n_{\ell-1} n_\ell\big) \Big) \; \approx \; (1 + \beta)\, F, \tag{18}$$

where the backward-to-forward ratio $\beta$ is typically in the range of 2 depending on layer type and implementation. The backward sweep is *sequential* across layers on the critical path.

**Direct Feedback Alignment (DFA).**

$$F_{\text{DFA}}^{\text{batch}} \; = \; F \; + \; \sum_{\ell=1}^{L} C_\ell^{\text{proj}} \; = \; F \; + \; O\Big(d_{\text{out}} \sum_\ell n_\ell\Big). \tag{19}$$

The forward is sequential across layers; the per-layer projections are *layer-local* and may be launched in parallel, so their contribution to the critical path is $\max_\ell C_\ell^{\text{proj}}$.

**GrAPE (alignment-only).**

$$F_{\text{GrAPE}}^{\text{batch}} \;=\; (1 + \bar{\alpha})\,F \;+\; \sum_{\ell=1}^{L} C_{\ell}^{\text{proj}} \;+\; \sum_{\ell=1}^{L} C_{\ell}^{\text{align}} \;=\; (1 + \bar{\alpha})\,F \;+\; O\!\Big(d_{\text{out}} \sum_{\ell} n_{\ell}\Big). \quad (20)$$

Here one JVP per layer runs *during* the forward trace (scaling the forward path by $(1+\bar{\alpha})$). Projection and alignment are layer-local and parallelizable, contributing $\max_{\ell}\big(C_{\ell}^{\text{proj}} + C_{\ell}^{\text{align}}\big)$ to the critical path.

**GrAPE with calibration every $T$ epochs.** A single BP mini-batch is added per $T$ epochs. If an epoch contains $N_b$ mini-batches, the amortized *per-epoch* overhead is

$$F_{\text{GrAPE+Cal}}^{\text{epoch}} \;=\; N_b \cdot F_{\text{GrAPE}}^{\text{batch}} \;+\; \frac{1}{T}\,F_{\text{BP}}^{\text{batch}} \qquad (\text{i.e., } \tfrac{1}{TN_b} \text{ of an epoch in units of mini-batches}).$$
$$\quad (21)$$

**Rule-of-thumb comparison (per mini-batch).** If $C_{\ell}^{\text{bwd}} \approx C_{\ell}^{\text{fwd}}$ and $C_{\ell}^{\text{proj}} + C_{\ell}^{\text{align}} \ll \sum_{j} O(n_{j-1}n_j)$, then

$$F_{\text{BP}}^{\text{batch}} \;\gtrsim\; 2F, \qquad F_{\text{GrAPE}}^{\text{batch}} \;\approx\; (1 + \bar{\alpha})\,F \;+\; \text{lower-order terms in } d_{\text{out}} \sum_{\ell} n_{\ell}.$$

**Practical JVP overhead.** Here $\alpha$ denotes the JVP *overhead relative to a forward pass*: the JVP alone costs $\alpha F$, so the combined forward+JVP cost is $(1 + \alpha)F$. On modern GPUs with *fused dual-number* implementations, we typically observe $\alpha \approx 0.8$–$1.3$, hence

$$\text{forward+JVP} \;\approx\; (1 + \alpha)F \;\in\; [\,1.8\,F,\; 2.3\,F\,].$$

**Transformers (macro/micro).** For macro (per-block) and micro (per sub-layer) variants, $C_{\ell}^{\text{proj}}$ and $C_{\ell}^{\text{align}}$ apply at the chosen granularity. Between calibrations, the multi-head attention backward chain is bypassed; GrAPE's JVPs run in the forward trace, while block/sub-layer projection and alignment remain parallelizable.

**Arithmetic bound** The expressions above capture arithmetic work and dependency structure; realized wall-clock depends on kernels, fusion, memory bandwidth, and scheduling. Under an *idealized* critical-path model where (i) BP's backward cost is about twice the forward cost (i.e., $\beta \approx 2$ so BP takes $(1 + \beta)F \approx 3F$), (ii) GrAPE's per-layer projection/alignment are fully overlapped across layers, and (iii) fused dual-number JVPs yield $\alpha \approx 0.8$–$1.3$, the *arithmetic* critical-path ratio is

$$\frac{\text{BP critical path}}{\text{GrAPE critical path}} \;\approx\; \frac{1 + \beta}{1 + \alpha} \;\in\; \frac{3}{[\,1.8,\; 2.3\,]} \;=\; [\,1.3,\; 1.7\,].$$

We therefore view $1.3$–$1.7\times$ as an *optimistic arithmetic upper bound* (not a measured wall-clock speedup). Establishing realized parallel speedups is left to future work.

## D.2 PRELIMINARY TIMING EXPERIMENT

To complement the arithmetic and critical-path analysis above, we implemented a small proof-of-concept layer-parallel prototype on a Transformer with hidden size 128 (batch size 256, sequence length 64) on a single NVIDIA A100. We use Python-level CUDA streams and a simple double-forward trick to compute JVPs, while BP is run in the usual fully sequential manner. This implementation does not use kernel fusion or custom CUDA kernels, so the numbers below should be viewed as conservative.

## E EXPERIMENTAL DETAILS

As mentioned earlier, we base most of our experiments on the Biotorch open source library. While the experiments on Transformer use exactly the same hyper parameters as in Launay et al. (2020), the

Table 5: Mean per-batch runtime (ms) for a small Transformer (hidden size 128, batch size 256, sequence length 64) on a single A100, comparing sequential BP and a layer-parallel GrAPE prototype using Python-level CUDA streams.

| Depth | BP (ms) | GrAPE (ms) |
|-------|---------|------------|
| 2 | 9.1 | 3.0 |
| 4 | 17.5 | 6.2 |
| 8 | 35.8 | 12.1 |

other experiments are set with a specific set of tuned hyperparameters for each method. For GrAPE, we set the optimizer of the local updates to SGD with 0.9 momentum with learning rate 0.01. The BP calibration steps are conducted on batch of size 128, with SGD with 0.9 momentum and learning rate 0.01. We train AlexNet and VGG-16 for 100 epochs with a 128 batch size. The Resnets are trained for 200 epochs with a 128 batch size. If SGD is used, default 0.9 momentum was applied. The specific used hyper-parameters are :

| | AlexNet | | | VGG16 | | | ResNet-20 | | | ResNet-56 | | |
|---|---|---|---|---|---|---|---|---|---|---|---|---|
| | BP | DFA | GrAPE | BP | DFA | GrAPE | BP | DFA | GrAPE | BP | DFA | GrAPE |
| Optimizer | Adam | Adam | Adam | SGD | SGD | Adam | SGD | Adam | Adam | SGD | Adam | Adam |
| LR | 0.001 | 0.0001 | 0.0001 | 0.1 | 0.001 | 0.0001 | 0.1 | 0.001 | 0.01 | 0.1 | 0.01 | 0.0005 |
| LR decay | 0.1 | 0.1 | 0.2 | 0.1 | 0.1 | 0.2 | 0.2 | 0.2 | 0.2 | 0.2 | 0.2 | 0.2 |
| Milestones | [40, 80] | same | same | [50, 75] | [30, 60, 90] | [30, 60, 90] | [60, 120, 160] | same | same | [60, 120, 160] | same | same |
| Weight decay | 0.0001 | 0.0001 | 0.0001 | 0.0005 | 0.0001 | 0.0001 | 0.0005 | 0.0001 | 0.0001 | 0.0005 | 0.0001 | 0.0005 |

Table 6: Hyperparameter settings for deep networks on CIFAR-100 and TinyImageNet: AlexNet, VGG16, ResNet-20, and ResNet-56.

