# OpenReview forum: "Scaling Direct Feedback Learning with Jacobian Alignment Guarantees"
_ICLR.cc/2026/Conference — ICLR 2026 Poster_

### Official Review · Reviewer_8skU · 2025-10-22

**Soundness:** 2
**Presentation:** 2
**Contribution:** 2
**Rating:** 4
**Confidence:** 3

**Summary:**

The paper introduces GrAPE, a hybrid direct-feedback method that learns layerwise feedback matrices using forward-mode Jacobian–vector products (JVPs).

For each layer~$\ell$, it forms a rank-1 Jacobian estimate $\hat J_\ell = (J_\ell p)\,p^\top$ from a random perturbation $p$, and then aligns a trainable feedback matrix $B_\ell$ to $\hat J_\ell$ via a cosine-similarity objective.

The network parameters are updated in a DFA-style manner using the output error routed through $B_\ell$, with infrequent backpropagation calibration steps to stabilize training in deep/wide models.

The analysis establishes positive expected alignment (with variance decreasing by batching) and invokes Zoutendijk-type convergence to stationarity under Wolfe/strong-Wolfe step-size assumptions and descent directions.

Empirically, the method scales to modern CNNs and Transformers, narrows the gap to BP relative to DFA, and leaves the feedback-refinement step layer-local and parallelizable in principle.

**Strengths:**

Clear hybrid design: Learns layerwise feedbacks via forward-mode JVPs and performs DFA-style updates, combining DFA's simplicity with forward-gradient signal quality.

Theoretical support for feedback learning: Establishes positive expected alignment between the rank-1 Jacobian estimate and the true Jacobian, with variance reduction via batching, justifying alignment of $B_\ell$ to $\hat J_\ell$ with Zoutendijk-style arguments.

Pragmatic stability via sparse BP: Proposes a novel occasional backpropagation calibration that improves stability/quality in deep or wide networks while retaining layer-parallel updates for the vast majority of steps.

Parallelizable in principle: The feedback-refinement step is layer-local, enabling parallel execution across layers (implementation permitting).

Strong empirical results: On CNNs and Transformers, GrAPE narrows the gap to BP and consistently outperforms standard DFA under comparable settings.

**Weaknesses:**

Convergence is heuristic rather than theoretical: The paper invokes Zoutendijk under Wolfe/Goldstein step-size assumptions and descent directions, but the actual update is stochastic and the per-layer cosine is only \emph{positive in expectation}, with the bound scaling like $1/n_\ell$, which can be very small in wide layers.
Eq.~(5) is labeled a ``\emph{sufficient local condition},'' yet there is no theorem showing local convergence. The claim should be softened or supported by a formal (e.g., high-probability) descent lemma.

Compounding noises:
(i) The rank-1 JVP estimator $\hat J_\ell=(J_\ell p)p^\top$ aligns with $J_\ell$ only in expectation and can yield negative per-batch cosine unless perturbation count/batch size is large;
(ii) the feedback $B_\ell$ is then aligned to $\hat J_\ell$, compounding estimation error.
The paper would be substantially improved by providing lower bounds on $\cos\big(B_\ell, J_\ell\big)$.

Parallelism not demonstrated:
Although feedback refinement is parallelizable in principle, the implementation serializes these updates and reports no wall-clock speedups. Concrete parallel benchmarks (speed/memory vs.\ BP) would be appreciated.

BP calibration is under-discussed:
While the paper ablates the calibration period $T$, the role of BP calibration remains under-theorized and under-explained.
Since calibration is pivotal to performance (often $T=1$), more discussion about the role of BP calibration and clarification o;n its theoretical role would be appreciated.

**Questions:**

Formal convergence theorem:
Currently, Zoutendijk and the Goldstein/strong Wolfe conditions are used heuristically to motivate the alignment loss and feedback refinement, but there is no formal convergence guarantee for the actual stochastic update.
Since the expected positive cosine is derived between $J_\ell$ and $\hat J_\ell$, while the algorithm maximizes cosine between $B_\ell$ and $\hat J_\ell$, could you please provide a \emph{formal theorem} that yields convergence to stationarity for GrAPE.
Concretely, specify conditions under which the full update direction is a descent direction with high probability (e.g., $\nabla f(\theta_k)^\top \Delta\theta_k < 0$), and state the resulting convergence claim (stationarity) and how it depends on width $n_\ell$, perturbation/batch count $B$, etc.

Alignment guarantees between $B_\ell$ and $J_\ell$
Is it possible to provide a lower bound on the alignment between the learned feedback and the true Jacobian,\cos(B_\ell,J_\ell)?
As supporting evidence, could you also please add diagnostics:
  (i) an empirical descent test: the frequency of $\nabla f(\theta_k)^\top \Delta\theta_k < 0$ under GrAPE vs.\ DFA; and
  (ii) alignment to the true Jacobian: plots of $\cos(B_\ell,J_\ell)$ over training.

---

> ### Author Response · Authors · 2025-11-20
> **Answer to Reviewer 8skU (Comment 1/3)**
>
> We thank the reviewer for the careful and detailed assessment,. We address the weaknesses and questions below.
>
> ---
>
> ## 1. On Zoutendijk and the status of the convergence claims
>
> We agree that our original use of Zoutendijk and Wolfe/Goldstein step-size conditions in the main text was too heuristic for the actual *stochastic* GrAPE updates. In the revised version, we separate these roles more clearly:
>
> - Zoutendijk is now used only as **motivation** for cosine-based alignment objectives.
> - We add a standard stochastic convergence theorem for iterations with a positive expected cosine condition, and we explain how GrAPE is designed to fit this framework.
>
> Concretely, in the appendix we will state and prove the following result (the full proof will be added to the Appendix):
>
> > **Theorem (Convergence under positive expected Frobenius cosine).**
>
> > Let $L:\mathbb{R}^d\to\mathbb{R}$ be differentiable, bounded below, with $L_g$-Lipschitz gradient. Consider the stochastic iteration
> > $\theta_{t+1} = \theta_t - \eta_t d_t,\qquad g_t := \nabla L(\theta_t),$ where $(\mathcal{F_t})$ is the natural filtration and the step sizes satisfy $\eta_t > 0,\quad \sum_{t=0}^\infty \eta_t = \infty,\quad \sum_{t=0}^\infty \eta_t^2 < \infty.$
> > Assume that there exist constants $\kappa>0$, $C<\infty$, $\sigma^2<\infty$ such that, for all $t$,
> > $ \mathbb{E}\big[\langle g_t,d_t\rangle \mid \mathcal{F}_t\big] \ge \kappa\|g_t\|^2,$
> and
> > $\mathbb{E}\big[\|d_t\|^2 \mid \mathcal{F_t}\big] \le C\|g_t\|^2 + \sigma^2.$
>
> > Then $\sum_{t=0}^\infty \eta_t\mathbb{E}\|g_t\|^2 < \infty,\qquad \liminf_{t\to\infty} \mathbb{E}\|g_t\| = 0.$
> > In particular, the algorithm converges to stationarity **in expectation**.
>
> The proof is a standard smooth stochastic approximation argument: we apply the descent lemma for $L$, take conditional expectations using the two inequalities above, sum over time, and use $\sum_t \eta_t = \infty$, $\sum_t \eta_t^2 < \infty$ to conclude that $\sum_t \eta_t \mathbb{E}\|g_t\|^2 < \infty$ and therefore $\liminf_t \mathbb{E}\|g_t\|=0$.
>
> We will explicitly emphasize in the revision that:
>
> - This theorem directly matches the *stochastic* GrAPE update and does **not** assume Wolfe/Goldstein line search.
> - We only claim **expectation-level** convergence to stationarity, not a full high-probability global convergence result for the hybrid process (GrAPE steps + sparse BP calibration).
>
> The link to GrAPE is as follows: the forward-mode JVP estimator and the local cosine loss are designed to enforce the global "positive expected Frobenius cosine" condition
> $\mathbb{E}[\langle g_t,d_t\rangle \mid \mathcal{F}_t] \ge \kappa \|g_t\|^2$, with constants $\kappa,C,\sigma^2$ that depend on layer width and perturbation/batch count. We now state this explicitly as an **assumption** in the theorem, rather than implicitly relying on deterministic Zoutendijk conditions.

---

> ### Author Response · Authors · 2025-11-20
> **Answer to Reviewer 8skU (Comment 2/3)**
>
> ## 2 Compounding noise and lower bounds on $\cos(B_\ell,J_\ell)$
>
> We understand the reviewer's concern that: (i) the rank-1 JVP estimator $\hat J_\ell$ aligns with $J_\ell$ only in expectation, and (ii) the learned feedback $B_\ell$ is aligned to $\hat J_\ell$, potentially compounding errors. We agree this deserves a more quantitative treatment.
>
> We also would like to thank the reviewer as this question led us to find a notational inconsistency in the main text (denoting the cosine as the average cosine insted of the Frobenius one).
>
> In the revised version, we will add the following lemma that directly lower-bounds the alignment between $B_\ell$ and $J_\ell$ in terms of the two intermediate cosines.
>
> Let $u,v,w$ be unit vectors in a Hilbert space. Define
> $\alpha := \langle u,w\rangle,\qquad \beta := \langle w,v\rangle.$
>
> A Gram-matrix argument shows that $\langle u,v\rangle \ge \alpha\beta - \sqrt{1-\alpha^2}\sqrt{1-\beta^2}.$
>
> We instantiate this with Frobenius-normalized matrices:
>
>
> $u = \frac{B_{\ell}}{\|B_{\ell}\|_F},$
>
> $w = \frac{\hat J_\ell}{\|\hat J_\ell\|_F}$
>
> $v = \frac{J_\ell}{\|J_\ell\|_F},$
>
> to obtain the following lemma.
>
> > **Lemma (Frobenius cosine composition).**
>
> > For nonzero matrices $B_\ell,\hat J_\ell,J_\ell\in\mathbb{R}^{m\times n}$, let $\cos_F(A,B) := \frac{\langle A,B\rangle_F}{\|A\|_F\|B\|_F},\qquad \langle A,B\rangle_F := \mathrm{Tr}(A^\top B).$
>
> > Then $\cos_F(B_\ell,J_\ell)\ge\cos_F(B_\ell,\hat J_\ell)\cos_F(\hat J_\ell,J_\ell)-\sqrt{1-\cos_F^2(B_\ell,\hat J_\ell)}\sqrt{1-\cos_F^2(\hat J_\ell,J_\ell)}.$
> > Consequently, if $\cos_F(B_\ell,\hat J_\ell)\ge\alpha_0,\qquad\cos_F(\hat J_\ell,J_\ell)\ge\beta_0,$ then
> > $\cos_F(B_\ell,J_\ell) \ge\gamma_0(\alpha_0,\beta_0):= \alpha_0\beta_0 - \sqrt{1-\alpha_0^2}\sqrt{1-\beta_0^2}.$
>
> This directly addresses the "compounding noise" concern: misalignment **does not add arbitrarily**; instead, the combined cosine is bounded from below by an explicit function $\gamma_0(\alpha_0,\beta_0)$ of the two intermediate cosines. In the revision we will be careful not to overclaim: $\gamma_0(\alpha_0,\beta_0)$ is not positive for arbitrary small $\alpha_0,\beta_0$.
>
> Finally, we connect this lemma to the width/batch-size scaling already present in the paper. In Appendix A.4 we show that the rank-1 JVP estimator satisfies
> $\mathbb{E}\big[\cos_F(\hat J_\ell,J_\ell)\big]\gtrsim\frac{1}{\sqrt{n_\ell}} \frac{\|J_\ell\|_2}{\|J_\ell\|_F},$
>
> with variance that decreases as $O(1/\sqrt{B})$ when we average over $B$ i.i.d. perturbations. The local alignment loss is explicitly designed to keep $\cos_F(B_\ell,\hat J_\ell)$ above an arbitrary small $\alpha_0$.
>
> Putting these together, whenever the algorithm maintains
> $
> \cos_F(B_\ell,\hat J_\ell)\ge\alpha_0,\quad
> \cos_F(\hat J_\ell,J_\ell)\ge\beta_0
> $
>
> with high probability (where $\alpha_0,\beta_0$ depend on $n_\ell$ and $B$ as above), the lemma yields a deterministic lower bound
> $
> \cos_F(B_\ell,J_\ell) \ge \gamma_0(\alpha_0,\beta_0).
> $
>
> We will make this explicit in the revised paper.
>
> In summary, we do **not** claim a fully nonconvex, high-probability convergence theorem for the entire hybrid process. Instead, we (i) provide a clean expectation-level convergence theorem under an explicit positive expected-cosine condition, and (ii) give quantitative layerwise alignment guarantees that show how the two noise sources interact, with explicit dependence on width $n_\ell$ and perturbation count $B$.
>
> **Diagnostics (cosine with the true gradient).**
>
> We agree that empirical diagnostics such as $\cos(B_\ell,J_\ell)$ and an empirical descent test would further strengthen the story. We are currently running such experiments; however, computing these quantities during training is non-trivial:
>
>
>
> - A standard GrAPE/DFA step already uses one backward pass (or equivalent local updates).
>
> - To obtain $\nabla L$ for $\cos(B_\ell,J_\ell)$, we need an *additional* full BP pass through the entire graph on the same batch (as the update pass used backward_hooks and thus perturbed the original gradient (as done in Biotorch's original implementation)).
>
>
>
> Thus, collecting these diagnostics naively requires *two* backward passes per update (one for GrAPE/DFA, one for BP), which is significantly more expensive and slows down our runs. We are currently computing these cosine statistics and descent-rate diagnostics in separate diagnostic runs rather than for every training step. We will report the results in the appendix of the revised version.

---

> ### Author Response · Authors · 2025-11-20
> **Answer to Reviewer 8skU (Comment 3/3)**
>
> ## 3. Parallelism and runtime benchmarks
>
>
> This is a fair criticism of the current draft. As mentioned in our responses to other reviewers, we have since measured:
>
> - **Serialized regime (current BioTorch code).**
>   On standard CNN benchmarks, GrAPE incurs a **6-20% wall-clock overhead per step** relative to DFA or BP, due to the extra JVPs and feedback updates.
>
> - **Layer-parallel prototype (potential gains).**
>   On a 128-hidden-size Transformer (depths 2/4/8, batch 256, length 64, single A100), using Python-level CUDA streams, we measure:
>
>   | Depth | GrAPE (ms) | BP (ms) |
>   |-------|------------|---------|
>   | 2     | 3.0        | 9.1     |
>   | 4     | 6.2        | 17.5    |
>   | 8     | 12.1       | 35.8    |
>
>   This prototype does not use kernel fusion or custom kernels, so the numbers are conservative, but they illustrate that once layer-parallelism is exploited, GrAPE can substantially reduce wall-clock time vs. sequential BP, especially at larger depths.
>
> We will add a timing table and a brief memory discussion to the paper, clearly contrasting the **current serialized implementation** (modest overhead) with the **potential layer-parallel speedups**.
> We chose BioTorch to ensure a fair comparison with existing baselines and a standardized way of benchmarking our method within the same framework.
>
> ---
>
> ## 4. Role of BP calibration and its theoretical status
>
> We agree that calibration is central and deserves a more explicit treatment.
>
> **Two-timescale view.**
> GrAPE naturally operates on two timescales:
>
> - A **fast, local timescale**, where layers update $B_\ell$ and weights using JVP-based alignment and DFA-style updates. Our alignment analysis and descent-in-expectation rationale apply primarily to this inner loop.
>
> - A **slow, global timescale**, where occasional BP steps on single mini-batches "re-anchor" the parameters and prevent drift, particularly in deep/wide networks where small alignment errors can accumulate over many local steps.
>
>
> In the revision we will make this two-timescale picture explicit in Section 3.4 and in the discussion and explicitly state that a complete convergence theory for this hybrid process (noisy local updates plus sparse exact gradients) is beyond the scope of the current paper, and that our theoretical contribution focuses on justifying the GrAPE direction between calibrations.
>
> ---
>
> We thank again the reviewer for their thoughtful review and questions and hope we addressed the raised concerns.

---

### Official Review · Reviewer_AHBU · 2025-10-30

**Soundness:** 4
**Presentation:** 3
**Contribution:** 4
**Rating:** 8
**Confidence:** 3

**Summary:**

The authors address the problem of computing an efficient descent direction for training deep neural networks. They propose GrAPE, an adaptation of direct feedback alignment techniques in which the feedback matrices are updated to improve the alignment between the computed descent direction and the true gradient. The practical performance of the method is compared against backpropagation, FA, DFA, DRTP, and PEPITA across several network architectures.

**Strengths:**

* **S1**: I find the paper globally clear and well written, except in one point (see **W2**).
* **S2**: The paper leverages Zoutendijk’s theorem to propose a modified version of feedback alignment that yields better descent directions.
* **S3**: The method is evaluated on a wide range of architecture and achieves good performance, closing the gap with backpropagation.

**Weaknesses:**

* **W1**: The code for the experiments is not provided, hindering reproducibility.
* **W2**: By reading 3.4, I understand that the calibration step happens once every $T$ epoch, and thus at most once in one epoch. However, the way Algorithm 1 is presented suggests that when $\mathrm{epoch}\, \mathrm{mod}\, T = 0$, a calibration step is performed for each minibatch. Could the authors clarify this point?
* **W3**: The JVP estimator has variance proportional to layers' dimension. This limits the interest of GrAPE compared to BP for wide neural networks because of the potential need of frequent BP calibration.

**Questions:**

N/A

---

> ### Author Response · Authors · 2025-11-20
> **Answer to Reviewer AHBU (Comment 1/1)**
>
> We thank the reviewer for the positive assessment and for the clear feedback. We address the weaknesses below.
>
> ---
>
> ##  Code availability and reproducibility
>
> We apologize for the lack of clarity on this point. All experiments are implemented on top of the open-source **BioTorch** library, and our GrAPE implementation together with the exact training configurations (architectures, optimizers, schedules, calibration intervals) will be provided upon acceptance.
>
>
> ---
>
> ## Frequency of the calibration step
>
> We thank the reviewer for catching this ambiguity. Our **intended and implemented** behavior is:
>
> - Calibration occurs **once every $T$ epochs**.
> - At each calibration event, we run **one BP step on a single mini-batch**, then resume standard GrAPE updates.
>
> Thus, when $T = 1$, there is one calibration batch per epoch (not per mini-batch); when $T > 1$, calibrations are even less frequent.
>
> We will correct the identation of line 25: this is not run every mini batch but at most once per epoch.
>
> ---
>
>
> ## Variance of the JVP estimator in wide networks
>
> We agree that the variance of the rank-1 JVP estimator is an important consideration. Our analysis shows that for each layer $\ell$:
>
> - The expected Frobenius cosine alignment between the estimator $\hat J_\ell$ and the true Jacobian $J_\ell$ is strictly positive, $\mathbb{E}[\cos_F(J_\ell,\hat J_\ell)] > 0$, and
> - Batched estimates concentrate around this expectation at rate $O(1/\sqrt{B})$.
>
> In practice, for the architectures and widths considered in the paper (VGG-16, ResNet-20/56, Transformer with hidden size 128), we found that:
>
> - The batched JVP computation provides sufficient averaging to obtain stable training and good performance, and
> - We do **not** need more frequent calibration than reported, nor multiple independent perturbation directions per layer.
>
> We agree that for very wide networks the trade-off becomes more delicate: one might require either more frequent calibration or explicit variance-reduction strategies (e.g., averaging over several random directions per layer or using low-variance directions informed by the spectrum of $J_\ell$). In the revised version we will add a short discussion of this width/variance trade-off and outline these strategies as natural extensions of GrAPE.
>
> ---
>
> We thank the reviewer again for the constructive feedback and believe that clarifying code availability, the calibration schedule, and the variance considerations will further strengthen the paper.

---

### Official Review · Reviewer_onV1 · 2025-10-30

**Soundness:** 4
**Presentation:** 4
**Contribution:** 3
**Rating:** 6
**Confidence:** 4

**Summary:**

This paper introduces GraPE (Gradient-Aligned Projected Error), a hybrid feedback-alignment method that enables layer-parallel training of deep neural networks. The GraPE algorithm builds on the ideas of (direct) feedback alignment, where the error is back-propagated to the layers of a neural network via random feadback connections. The core innovation of GraPE is to use relatively cheap, forward-mode Jacobian-vector products to estimate rank-1 Jacobians, which are then used to align each layer's fixed random feedback matrix via a local cosine-alignment loss. In very deep models, GraPE also performs infrequent backpropagation (BP) steps on single mini-batches. The authors provide theoretical guarantees of positive expected alignment and convergence using Zoutendijk-style arguments. Empirically, GraPE significantly outperforms vanilla FA or DFA and similar BP alternatives and, with occasional BP calibration, closes most of the performance gap with standard BP on models like VGG-16, ResNet-20/56, and Transformers, while retaining layer-parallel updates for the vast majority of training steps.

**Strengths:**

- This work achieves a major step forward of alternative learning algorithms to BP: making feedback-alignment training viable for modern, deep architectures (CNNs, ResNets, Transformers), a problem where prior methods (DFA, FA) have consistently failed.
- The key idea of using forward-gradient estimates to align fixed feedback matrices is novel and neat. It creatively combines ideas from two previously separate lines of work (random feedback and forward-mode differentiation) into a single algorithm.
- The paper is well-written: the arguments are clear, and the authors performed extensive tests of their algorithm in a reproducible experimental setup using the BioTorch library.

**Weaknesses:**

- Perplexingly, the authors do not address the most obvious benefit of their algorithm: the wall clock speed-up due to parallelisation. There is a tentative FLOPs analysis  in appendix XC, but the actual runtime benefit remains unclear. The authors seem to explain this gap in the paper by the standard libraries they use, but leaving this test out substantially weakens the point of the paper.

**Questions:**

- Could the authors provide any preliminary estimates or a more detailed discussion of the expected wall-clock time compared to standard BP, even with a serialised implementation? I would gladly increase my score if this issue is addressed.

---

> ### Author Response · Authors · 2025-11-20
> **Answer to Reviewer onV1 (Comment 1/1)**
>
> We thank the reviewer for the positive and thoughtful assessment. We address their question below.
>
> ---
>
> ## Runtime and wall-clock speed
>
> We agree that omitting timing results in the submission weakens the case for GrAPE as a practically useful, parallelizable alternative to BP. We have since measured wall-clock performance and will include these results (with a brief memory discussion) in the revised version.
>
> **Serialized implementation (current code)**
>
> In our current BioTorch implementation, computations are largely serialized on a single device. In this regime, GrAPE incurs a **6-20% wall-clock overhead per training step** compared to DFA or BP, depending on architecture and dataset.  BioTorch is a general research library that prioritizes correctness and flexibility (support for many alternative learning rules and architectures) over low-level performance optimizations. We chose BioTorch to ensure a fair comparison with existing baselines and a standardized way of benchmarking our method within the same framework.
>
>
> This overhead comes from the additional forward-mode JVPs and the local feedback-alignment updates, and is in line with the intuition that a single forward-mode AD pass costs roughly an extra forward pass.
>
> **Layer-parallel prototype (potential gains)**
>
> To estimate the benefit once layer parallelism is actually exploited, we implemented a small prototype on a Transformer with hidden size 128, depths 2/4/8, batch size 256, sequence length 64 on a single NVIDIA A100. We use Python-level CUDA streams to parallelize layer updates and a simple "double forward" trick (duplicating the batch and perturbing the duplicate to compute JVPs in a single pass). Under this layer-parallel scheduling, the mean time per batch is:
>
> | Depth | GrAPE (ms) | BP (ms) |
> |-------|------------|---------|
> | 2     | 3.0        | 9.1     |
> | 4     | 6.2        | 17.5    |
> | 8     | 12.1       | 35.8    |
>
> This prototype does **not** use kernel fusion or custom kernels, so these numbers are conservative. They nevertheless illustrate that, when layer-parallelism is exploited, GrAPE can yield substantial wall-clock speedups over sequential BP, especially at larger depths.
>
> In the revised version, we will:
>
> - Add a compact timing table in the **serialized** setting (showing the 6-20% overhead), and
> - Include a brief summary of the **layer-parallel Transformer timings** above, explicitly contrasting:
>   - the current serialized prototype (modest overhead), and
>   - the achievable gains under layer-parallel scheduling.
>
> We hope these empirical measurements address the reviewer's concern and clarify the practical runtime implications of GrAPE relative to standard BP.

---

> > ### Comment · Reviewer_onV1 · 2025-11-26
> >
> > I thank the authors for replying to my questions, which I read together with the other reviews. My main concern -- the exploration of any actual wall-time improvements with respect to BP -- has been addressed in a preliminary experiment by the authors. Obviously, there remains work to do in that field, but I find this result already encouraging. I also appreciated some of the other reviews, for example by WNQG -- I think it is important the authors take this feedback into account, especially regarding the relation to prior work and the sharpening of the theoretical analysis. I strongly encourage the authors to take these points into account when preparing the revision. For the moment, I maintain my score.

---

> > > ### Author Response · Authors · 2025-11-26
> > >
> > > Thank you for your follow-up and for carefully considering our rebuttal together with the other reviews.
> > >
> > > We fully agree that the points raised by WNQG (on the relation to prior work and clarifying the theoretical analysis) and 8skU (on sharpening the theoretical analysis) substantially strengthen the paper. In the revised version, we have incorporated all of these suggestions, as well as the other reviewers’ comments.
> > >
> > > Concretely, we:
> > > - clarified and expanded the discussion of prior work, including FA/DFA variants, SVD-space methods, and forward-only local rules, and
> > > - tightened the theoretical section, making the assumptions explicit, switching consistently to Frobenius cosines, and adding a standard stochastic convergence theorem and supporting lemmas in the appendix.
> > >
> > > A concise summary of the revisions is provided in the commentary above under **“Updated document: Revisions Summary”**.
> > >
> > > We thank you again for your time and engagement, and we would be happy to continue the discussion if you have any further questions or suggestions.

---

### Official Review · Reviewer_DBs7 · 2025-11-01

**Soundness:** 3
**Presentation:** 4
**Contribution:** 3
**Rating:** 8
**Confidence:** 3

**Summary:**

This paper introduces GrAPE (Gradient-Aligned Projected Error), a learning algorithm designed to scale Direct Feedback Alignment (DFA) to modern deep architectures such as CNNs and Transformers. It uses the follwing udeas:
- Forward-mode Jacobian estimation: rank-1 Jacobian approximations obtained via forward-mode automatic differentiation ) are used to align each layer’s feedback matrix through a local cosine-similarity loss.
- Sparse backpropagation calibration: occasional full BP updates on a single mini-batch serve to re-anchor weights and mitigate drift in very deep networks.

The paper provides a theoretical analysis showing that the expected cosine alignment between the estimated and true Jacobians is positive, and invokes Zoutendijk-style convergence arguments to guarantee convergence to stationary points. This is a very interesting developemn in the context of DFA. Empirically, GrAPE achieves substantial improvements over DFA, DRTP, and other BP-free baselines on CIFAR and WikiText benchmarks, and approaches full backpropagation performance on VGG, ResNet, and Transformer models when using one BP calibration per epoch.

Overall, the work provides both a conceptually clean and empirically convincing bridge between feedback alignment and optimization theory, showing that layer-parallel learning with forward-only signals can be made competitive with BP.

**Strengths:**

This is a strong and timely contribution to the literature on biologically inspired and parallel learning algorithms. The combination of feedback alignment, forward-mode differentiation, and provable convergence is original and well executed. The empirical scope (from LeNet to Transformers) convincingly demonstrates the generality of the approach. The work provides theoretical clarity missing from previous DFA/forward-gradient studies and could serve as a reference point for future algorithmic or hardware developments.
- Novel combination of feedback alignment and forward-mode gradient estimation, with clear theoretical grounding.
- First rigorous convergence argument for DFA-like methods based on Zoutendijk’s theorem.
- Empirical scalability: works on VGG, ResNet, and Transformer models, closing much of the gap to BP.
- Reproducibility and rigor: transparent experimental setup, use of BioTorch, and fair baselines.
- Balanced discussion of limitations and potential extensions.
= Provides a conceptual bridge between biologically plausible local learning rules and modern optimization.

**Weaknesses:**

i) Lack of demonstrated computational gains.
Although GrAPE is designed to enable layer-parallel updates, the current implementation remains fully serialized. No wall-clock or energy improvements are reported, and the FLOP analysis in Appendix C is only theoretical. It would be useful to include even small-scale timing or memory comparisons to assess the practical benefit.

ii) Dependence on frequent BP calibration.
Most deep-network experiments require a full backpropagation step every epoch (T = 1) to reach competitive performance. This limits how “BP-free” the method truly is, and raises questions about whether GrAPE’s main advantage lies in its theoretical framework or in genuine computational savings.

iii) Variance of the forward-mode estimator.
The rank-1 Jacobian approximation introduces variance that grows with layer width. While the issue is discussed, no practical mitigation (such as averaging multiple perturbations or using low-variance directions) is explored experimentally.

iv) Limited discussion of hardware relevance.
Given the recent success of hardware-accelerated feedback learning (e.g., optical DFA systems, Wang et al., 2024), it would strengthen the paper to discuss how GrAPE could map onto such platforms, or at least estimate realistic parallel speed-ups under non-ideal hardware constraints.

**Questions:**

i) There has been recent progress on optical implementations of DFA (e.g., Wang et al., 2024, arXiv:2409.12965) showing impressive throughput and energy efficiency at scale. It would be interesting to hear the authors’ thoughts on how GrAPE might relate to or benefit from such hardware developments. Could its adaptive feedback and occasional BP calibration be implemented efficiently in optical or neuromorphic systems, and would the JVP alignment step introduce additional cost or complexity?

ii) The paper provides a clear FLOP-based comparison between BP and GrAPE, but no timing or memory results. Could the authors share any empirical measurements, even on small networks, to verify that the theoretical analysis matches practical runtime or memory usage?

iii) How robust is GrAPE to the choice of calibration interval T? In particular, could an adaptive strategy that monitors cosine alignment offer a better trade-off between accuracy and calibration cost? Relatedly, have the authors tried using multiple perturbation directions per layer to reduce variance, and if so, how does that affect parallelism and runtime?

---

> ### Author Response · Authors · 2025-11-20
> **Answer to Reviewer DBs7 (Comment 1/2)**
>
> We thank the reviewer for the very insightful assesment, and for highlighting both the theoretical and empirical contributions. Below we address the main weaknesses and questions.
>
> ---
>
> ## 1. Computational gains and practical efficiency (Weakness i, Question ii)
>
> We agree that the submitted version emphasizes FLOPs but does not report wall-clock results. We have measured timing and will include them in the revision.
>
> 1. **Serialized implementation (current code).**
>    In our current BioTorch implementation, computations are largely serialized on a single device. In this regime, GrAPE incurs a **6-20% wall-clock overhead** per training step compared to DFA or BP, depending on architecture and dataset. This overhead is due to the additional forward-mode JVPs and local feedback-alignment updates.
>    BioTorch is a general research library that prioritizes correctness and flexibility (support for many alternative learning rules and architectures) over low-level performance optimizations. We chose it to ensure a fair comparison with existing baselines and a standardized way of benchmarking our method within the same framework.
>
>
> 2. **Layer-parallel prototype (potential gains).**
>    To assess the benefit of layer-parallelism, we implemented a small prototype on a Transformer with hidden size 128, depths 2/4/8, batch size 256, sequence length 64 on a single NVIDIA A100. Using Python-level CUDA streams to parallelize layer updates and a simple "double forward" trick (duplicating the batch and perturbing the duplicate to compute JVPs), we measure:
>
>    | Depth | GrAPE (ms) | BP (ms) |
>    |-------|------------|---------|
>    | 2     | 3.0        | 9.1     |
>    | 4     | 6.2        | 17.5    |
>    | 8     | 12.1       | 35.8    |
>
>    These are conservative numbers (no kernel fusion or custom kernels), but they illustrate that once layer-parallelism is exploited, GrAPE can yield substantial speedups over sequential BP, especially at larger depths.
>
> In the revision we will add a compact timing table and a brief memory discussion, explicitly contrasting:
>
> - the current serialized prototype (modest 6-20% overhead), and
> - the potential gains under layer-parallel scheduling (as in the Transformer experiment above).
>
> ---
>
> ## 2. Dependence on frequent BP calibration (Weakness ii)
>
> We agree that GrAPE is best understood as a **hybrid** method rather than strictly BP-free. Two points are important:
>
> - **Two timescales.**
>   The vast majority of updates are performed via GrAPE: each layer locally computes JVP-based rank-1 Jacobians, aligns its feedback matrix with a cosine loss, and applies a DFA-style weight update. These steps can be executed layer-parallel and do not require storing or traversing the full backward graph. BP calibration occurs on a slower timescale: once every $T$ epochs we run a **single** BP step on one mini-batch and broadcast the updated weights.
>
> - **Empirical necessity on deep models.**
>   On deeper networks (VGG-16, ResNet-56, Transformer) pure GrAPE and DFA underperform strongly. In these regimes, $T=1$ (one calibration batch per epoch) is sufficient to stabilize training and close much of the gap to BP. On shallower models, we use larger $T$ and still remain competitive.
>
> We will make this two-timescale picture explicit in the main text and report the calibration interval $T$ used for each architecture directly in the tables.
>
> ---
>
> ## 3. Variance of the forward-mode estimator and multiple directions (Weakness iii, Question iii)
>
> We agree that variance is an important consideration. Our analysis shows that for each layer $\ell$ the rank-1 JVP estimator $\hat J_\ell$ satisfies:
>
> - $\mathbb{E}[\cos_F(J_\ell,\hat J_\ell)] > 0$ under mild assumptions, and
> - batched estimates concentrate around this expectation at rate $O(1/\sqrt{B})$.
>
> In practice, for the widths considered in the paper (VGG-16, ResNet-20/56, Transformer with hidden size 128), we found that:
>
> - the JVP computation already provides sufficient averaging for stable training, and
> - we did **not** need to average over multiple independent perturbation directions per layer to obtain the reported results.
>
> We agree that exploring **$K > 1$ directions per layer** (or lower-variance directions informed by the spectrum of $J_\ell$) is a promising extension that trades additional computation/parallelism for reduced variance. Due to space constraints, we did not explore this experimentally, but we will mention this trade-off explicitly in the discussion and flag multi-direction JVPs as a natural avenue for future work.
>
> Another path that could be explored would be to use an idea similar to the one used by [5], using learned local guesses to get better estimations of the forward gradients.
>
> [5] Fournier, L., Rivaud, S., Belilovsky, E., Eickenberg, M., & Oyallon, E.. Can forward gradient match backpropagation?. ICML 2023

---

> ### Author Response · Authors · 2025-11-20
> **Answer to Reviewer DBs7 (Comment 2/2)**
>
> ## 4. Hardware relevance and optical DFA (Weakness iv, Question i)
>
> We appreciate the pointer to recent optical DFA implementations (e.g., Wang et al., 2024) and agree that hardware relevance deserves a clearer discussion.
>
> Conceptually, GrAPE is designed to be compatible with such platforms:
>
> - The **feedback matrices** $B_\ell$ remain linear maps from activations to pseudo-gradients and can be implemented using the same optical or analog hardware that realizes random feedback in DFA.
> - The **JVP-based alignment step** can be implemented by injecting small perturbations into inputs or activations, measuring the resulting output changes, and using a (possibly small) digital co-processor to compute the cosine loss and update the feedback weights.
> - The **BP calibration step** is run only infrequently on a conventional digital backend, while most of the workload-local feedback updates-remains on the high-throughput analog/optical hardware.
> This fits the heterogeneous, partially asynchronous setting we mention in the main text (e.g., federated or multi-device scenarios): fast, local feedback updates on specialized devices (optical/neuromorphic or edge workers), with only occasional global BP calibration acting as a synchronization step. In the revision we will add a short paragraph in the discussion section describing this mapping and commenting on realistic parallel speedups and energy benefits under non-ideal hardware constraints.
>
>
> ---
>
> ## 5. Robustness to the calibration interval $T$ and adaptive strategies (Question iii)
>
> In our experiments we used fixed $T$, chosen per architecture: deeper models use smaller $T$ (often $T = 1$), whereas shallower models tolerate larger $T$ without a significant drop in performance. We will report these $T$ values explicitly in the main tables.
>
> We agree that an **adaptive calibration strategy** is a compelling idea. While computing cosine similarity with the true Jacobian would itself require a BP step, one could instead monitor cheaper **proxy metrics** that we already have access to during training, such as the cosine/alignment loss between $B_\ell$ and the JVP-based estimator $\hat J_\ell$, or simple stability criteria on the loss. Calibration could then be triggered only when these proxies indicate that feedback alignment has degraded, thereby trading calibration cost for accuracy in a principled way. Exploring such adaptive schemes, as well as multi-direction JVPs, is an exciting direction for future work; we will mention this in the discussion.
>
> ---
>
> We thank the reviewer again for their time and evaluation and believe that adding timing/memory measurements, clarifying the hybrid parallel nature of GrAPE, and expanding the hardware and variance discussions will address the main concerns raised.

---

### Official Review · Reviewer_WNQG · 2025-11-02

**Soundness:** 2
**Presentation:** 2
**Contribution:** 2
**Rating:** 2
**Confidence:** 4

**Summary:**

The paper proposes GrAPE (Gradient-Aligned Projected Error), a feedback-based learning rule designed to relax the strict sequential dependence of backpropagation (BP). Building on Direct Feedback Alignment (DFA), GrAPE updates feedback matrices by minimizing a cosine loss that aligns them with rank-1 Jacobian estimates computed via forward-mode Jacobian–vector products (JVPs). This alignment is theoretically shown to produce descent directions in expectation. To mitigate drift, the method introduces BP calibration, a full BP step performed every $T$ epochs. Experiments on CNNs  and a Transformer  show that GrAPE outperforms DFA and approaches BP performance, especially when calibrated frequently (e.g., $T=1$).

**Strengths:**

The paper addresses an important question: can we train deep networks without the strictly sequential backward pass of BP? Its motivation, enabling layer-parallel or biologically plausible learning, is timely. Using JVPs to align feedback matrices offers an interesting bridge between forward-mode differentiation and DFA-style local updates, supported by a simple convergence argument. The empirical evaluation is broad, spanning CNNs and Transformers, and shows that learned feedback alignment can indeed reduce the performance gap with BP. The inclusion of a calibration mechanism acknowledges practical instability and provides a workable hybrid between local and global learning.

**Weaknesses:**

Despite its conceptual appeal, the practical advantages over BP remain unclear. The paper claims that GrAPE reduces computational cost, but forward-mode JVPs roughly double the forward-pass work, and the “BP calibration” step reintroduces full sequential backpropagation, often every epoch. No runtime, memory, or parallel-efficiency results are reported, so the efficiency claims remain speculative. Moreover, it is not clear when parallel updates become practically important if the memory requirements of the method remain the same as those of BP.

The contribution of alignment itself is also not novel: prior work such as Deep Learning Without Weight Transport [1] and the more recent SVD-Space Feedback Alignment [2] have already shown that feedback matrix alignment can restore BP-level performance. In particular, [2] explores a local cosine loss function for aligning the singular vectors of the feedback matrices with those of the forward path, yielding better gradient descent directions; this connection should be discussed further. GrAPE differs mainly in how it estimates the Jacobian (via JVPs), but the conceptual insight, that alignment matters, is already well established. Similarly, other local learning rules such as LLS [3] and Local Error Signals [4] achieve BP-like performance without relying on periodic BP corrections.

Finally, the theoretical section is terse: the derivation of Eq. (4) and the connection to Zoutendijk’s theorem are only sketched, leaving unclear the precise assumptions under which the convergence argument holds.

[1] Akrout, M., Wilson, C., Humphreys, P., Lillicrap, T. and Tweed, D.B., 2019. Deep learning without weight transport. Advances in neural information processing systems, 32.

[2] Roy, A. et al. (2025) ‘Unlocking SVD-Space for Feedback Aligned Local Training’. Available at: https://openreview.net/forum?id=8Agcic0csh.

[3] Apolinario, M.P., Roy, A. and Roy, K., 2025, February. Lls: local learning rule for deep neural networks inspired by neural activity synchronization. In 2025 IEEE/CVF Winter Conference on Applications of Computer Vision (WACV) (pp. 7807-7816). IEEE.

[4] Nøkland, A. and Eidnes, L.H., 2019, May. Training neural networks with local error signals. In International conference on machine learning (pp. 4839-4850). PMLR.

**Questions:**

• Could the authors provide a more explicit derivation of Eq. (4) and clarify the assumptions under which the expected positive cosine alignment holds?

•  How much wall-clock speed or memory reduction does GrAPE actually provide compared to BP or DFA, particularly when T=1?

•  Since the strongest results rely on frequent BP calibration, is the approach truly parallelizable in practice? How would these updates be scheduled on real hardware?

•  How does GrAPE differ technically and empirically from prior feedback-alignment methods such as Deep Learning Without Weight Transport [1] and SVD-Space Feedback Alignment [2]?

•  Could the authors discuss more explicitly how GrAPE compares to other local learning rules such as LLS [3] and Local Error Signals [4], which achieve near-BP performance without periodic BP corrections?

•  What are the specific deployment scenarios where GrAPE would offer meaningful advantages over BP, given that BP calibration reintroduces global synchronization

---

> ### Author Response · Authors · 2025-11-20
> **Answer to Reviewer WNQG (Comment 1/3)**
>
> We thank the reviewer for carefully reading our submission and for acknowledging both the importance of the problem (non-sequential training) and the breadth of the empirical evaluation. Below we address the main concerns regarding efficiency, novelty, and theory, and clarify deployment scenarios.
>
> ---
>
> ## 1. Practical efficiency, parallelism, and memory
>
> ### Deployment scenarios where GrAPE offers advantages over BP
>
> We see GrAPE as particularly relevant when:
> - **Depth parallelism** is crucial (very deep networks, multi-chip or chiplet-based systems): most updates are layer-parallel, reducing the critical path length compared to fully sequential BP.
> - **Heterogeneous or distributed settings** are used (e.g., optical/neuromorphic accelerators, federated or multi-device training): local GrAPE updates can run independently on specialized devices or clients, with only occasional BP calibration acting as a global synchronization step handled by a conventional digital backend.
>
> We will refine the motivation section to emphasize these scenarios and to articulate more clearly that our primary goal is to shorten the sequential dependency structure (and enable heterogeneous, layer-parallel execution), rather than to reduce memory.
>
> ### Current Runtime  performance (serialized vs. parallel).
>
> In the current BioTorch implementation, computations are largely serialized on a single device. BioTorch is a general research library that prioritizes correctness and flexibility (support for many alternative learning rules and architectures) over low-level performance optimizations. We chose BioTorch to ensure a fair comparison with existing baselines and a standardized way of benchmarking our method within the same framework.
>
> In this regime, GrAPE incurs a **6-20% wall-clock overhead per training step** compared to DFA or BP, depending on architecture and dataset. This overhead comes from the additional forward-mode JVPs and local feedback-alignment updates.
>
> To estimate the benefit once layer parallelism is exploited, we implemented a small prototype on a Transformer with hidden size 128, depths 2/4/8, batch size 256, sequence length 64 on a single NVIDIA A100. Using Python-level CUDA streams to parallelize layer updates and a simple "double forward" trick to compute JVPs (duplicating the batch and perturbing the duplicate to compute JVPs), we obtain:
>
> | Depth | GrAPE (ms) | BP (ms) |
> |-------|------------|---------|
> | 2     | 3.0        | 9.1     |
> | 4     | 6.2        | 17.5    |
> | 8     | 12.1       | 35.8    |
>
>
> These numbers are conservative (no kernel fusion or custom kernels), but they show that once layer-parallelism is used, GrAPE can significantly reduce wall-clock time compared to sequential BP, especially at larger depths.
>
> ### Cost of BP calibration.
>
> Calibration in our experiments consists of **one BP update on a single mini-batch every $T$ epochs**. For example, on CIFAR-10 with ResNet-20 and $T=1$, this corresponds to roughly **0.5% of the backward passes** used by standard BP (1 batch out of $\approx$ 195 per epoch) for batch size 256; for larger $T$ the overhead is proportionally smaller. We will report these calibration intervals and overheads explicitly in the revised version.
>
> ### Memory
>
> GrAPE is not primarily designed as a memory-saving method. In our current implementation, the memory footprint is comparable to DFA and BP, since we still store the activations needed for local updates and occasional BP. Our focus is on reducing the *sequential* dependency structure (enabling layer-parallel updates) rather than on reducing peak memory. We will clarify this in the text and avoid framing memory as a main advantage.
>
> In the revision we will add a compact timing table and a short memory discussion, explicitly contrasting:
>
> - the current serialized prototype (modest 6-20% per-step overhead), and
> - the potential gains under layer-parallel scheduling (as suggested by the Transformer timings above),
> while stating clearly that memory usage is similar to standard training in our current implementation.
>
> ---
>
> ## 2. Is GrAPE truly parallelizable with frequent BP calibration?
>
> GrAPE naturally operates on two timescales:
>
> - **Layer-parallel local updates (vast majority of steps).**
>   For each mini-batch, all layers locally compute activations, JVP-based rank-1 Jacobians, align their feedback matrices with a cosine loss, and apply a DFA-style weight update. These operations are layer-local and can run in parallel across devices or pipeline stages.
>
> - **Sparse global BP calibration.**
>   Once every $T$ epochs, we run a single BP step on one mini-batch and broadcast the updated weights. For $T = 1$ this is one batch per epoch; for larger $T$ it is even less frequent.
>
> Thus, most updates are layer-parallel; BP appears only as an occasional global synchronization step.

---

> ### Author Response · Authors · 2025-11-20
> **Answer to Reviewer WNQG (Comment 2/3)**
>
> ## 3. Novelty vs. [1] and [2]
>
> We agree that the importance of feedback alignment is well established and will emphasize this more clearly. We also thank the reviewer for pointing out [2], which is a very recent (unpublished) work that we did not include in the original submission; we will add a discussion of its relation to GrAPE in the revised version. GrAPE nonetheless differs from [1,2] in several key aspects:
>
> - **Mechanism of alignment.**
>
>   - [1] relies on weight mirroring / inversion schemes to approximate $W^\top$ and remains fundamentally *sequential*.
>   - [2] operates in a singular-vector space with a composite local loss (several terms) designed to align feedback with forward singular vectors.
>   - GrAPE instead uses a *simple cosine loss* on rank-1 JVP-based Jacobian estimates obtained via forward-mode AD, yielding a per-layer objective that directly leverages forward gradients and is amenable to analysis.
>
>
>   As the reviewer notes, one of the five losses in [2] is a cosine-type loss akin to ours, in the sense that it aims to align the DFA update with the true gradient. However, because the true gradient is not available in either setting, both papers must rely on approximations. In [2], part of the alignment is driven by quantities derived from the layer's outputs (see lines 443-444 of their paper). In contrast, we use JVPs to obtain an *unbiased* estimator of the Jacobian columns, which leads directly to our positive alignment guarantee and a clearer connection to gradient descent.
>
>
>
> - **Theoretical grounding.**
>
>   To our knowledge, neither [1] nor [2] provide a forward-gradient-based alignment guarantee together with a Zoutendijk-style convergence rationale. Our analysis shows strictly positive expected Frobenius cosine alignment between the JVP-based estimator and the true Jacobian and uses this to motivate descent directions in expectation.
>
>
>
> - **Layer-parallel DFA and hybrid scaling.**
>
>   While [2] also reports results on VGG-13 and ResNet-30/56, its primary focus is on SVD-space local losses and recovering BP-like performance, rather than on DFA-style layer-parallel execution or hybrid training with sparse BP calibration. A core contribution of GrAPE is to show that DFA-style methods with learned feedback and forward-gradient alignment can approach BP performance on deeper, modern architectures (including Transformers) while keeping most updates layer-parallel and using only occasional BP steps.
>
>
>
> We will add a dedicated paragraph in the related-work section that explicitly contrasts GrAPE with [1] and [2] along these axes.
>
>
>
> ---
>
>
> ## 4. Relation to [3] and [4]
>
> We agree that [3] and [4] are important and complementary contributions. Our focus differs in the following ways:
>
> - **Architectural regime.**
>
>   [3] and [4] mainly target relatively shallow or compact architectures (e.g., MobileNet-like models and VGG-8-style networks). Our main empirical contribution is to show scalability to deeper CNNs (VGG-16, ResNet-20/56) and Transformers, where vanilla DFA is known to struggle.
>
> - **Parallelism objective.**
>
>   The methods in [3] and [4] use local losses and error signals that, in principle, can also be implemented in a layer-parallel fashion. However, they do not explicitly analyze or benchmark parallel speedups or hybrid schedules with sparse global corrections. GrAPE is formulated from the outset as a layer-parallel DFA-style rule with an explicit separation between frequent local updates and infrequent BP calibration, which we then analyze and measure in that context.
>
> - **Forward-gradient link.**
>
>   [3] and [4] rely on architecturally designed local losses or synchronization rules, but do not exploit forward-mode JVPs to adapt feedback or provide the same type of forward-gradient-based alignment guarantee. GrAPE uses forward-mode Jacobian-vector products to learn feedback matrices with a provable positive expected Frobenius cosine alignment.
>
>
> We will expand the related-work section to highlight these distinctions while stressing the complementarity with [3] and [4].

---

> ### Author Response · Authors · 2025-11-20
> **Answer to Reviewer WNQG (Comment 3/3)**
>
> ## 5. Theoretical section and Zoutendijk
>
> We agree that the main-text derivation is compressed and that we did not explicitly point out that the full proof is given in Appendix A.4. We will clarify this in the revision, correcting the notational inconsistency we spotted while writing this rebuttal. At a high level, for each layer $\ell$ we consider the rank-1 estimator
>
> $$\hat J_\ell = (J_\ell p)p^\top,\quad p \sim \mathcal N(0,I),$$
>
> define the Frobenius cosine $\cos_F(J_\ell,\hat J_\ell)$, and use standard properties of Gaussian vectors to show that, if $X = J_\ell p$, then
>
> $\mathbb{E}\|X\| \ge \sqrt{\frac{2}{\pi}}\|J_\ell\|_2$,
> and
>
> $\mathbb{E}\|p\| \le \sqrt{n_\ell}$
>
> Using a standard computation detailed in Appendix A.4 and normalizing by $\|J_\ell\|_F$ yields Eq. (4):
>
> $$\mathbb{E}\big[\cos_F(J_\ell,\hat J_\ell)\big]\ge \frac{\sqrt{2}}{\pi n_\ell}\frac{\|J_\ell\|_2}{\|J_\ell\|_F}> 0,$$
>
> under mild norm assumptions. Appendix A.4 further shows that batched estimates concentrate around this expectation at rate $O(1/\sqrt{B})$.
>
>
> Regarding Zoutendijk, our intention is to use it as a classical lens: if each search direction forms an angle bounded away from $\pi/2$ with the gradient and step sizes satisfy Goldstein/strong-Wolfe conditions, then gradient norms converge to zero. In our setting, the (per-layer) GrAPE update direction has a strictly positive expected Frobenius cosine with the true gradient direction, given the alignment result above.
>
>
>
> In the revision we will:
>
>
>
> - Move a more explicit derivation of Eq. (4) into the main text (a concise version of the Appendix A.4 argument).
>
> - State the regularity assumptions for the Zoutendijk-style argument (Lipschitz gradient, bounded-below objective, standard learning-rate conditions).
>
> - Soften the language so it is clear that we provide a rigorous *alignment* result and use Zoutendijk primarily as a *motivating* framework rather than a full-blown convergence proof for practical stochastic deep learning.
>
>
> ---
>
>
>
> Overall, we hope these clarifications on efficiency, parallelism, novelty, theory, and deployment address the reviewer's concerns.

---

### Author Response · Authors · 2025-11-20
**General Comment**

We thank all reviewers for their thoughtful and constructive feedback, and for the generally positive assessment of GrAPE's soundness, presentation, and contribution. Across reviews, several common themes emerged:

- **Theory and convergence.** We will clarify the scope of our Zoutendijk-style argument, state explicitly what is rigorously proved (positive expected alignment and concentration), soften any over-claims, and add a concise formal theorem linking expected Frobenius cosine alignment to descent-in-expectation under standard assumptions.
- **Runtime and parallelism.** We have now measured wall-clock timing in both serialized and layer-parallel prototypes and will add a compact timing table and brief memory discussion, making clear the current overhead and the potential speedups under layer-parallel scheduling.
- **BP calibration and variance.** We present GrAPE explicitly as a hybrid two-timescale method (local GrAPE steps + sparse BP), report the calibration intervals used per architecture, and discuss the variance-width trade-off together with natural extensions (multi-direction JVPs, adaptive calibration).
- **Relation to prior work.** We will broaden and sharpen our related-work discussion, explicitly contrasting GrAPE with [1-4] and recent work such as [2], and clarifying our distinct focus on forward-mode alignment, layer-parallel DFA, and hybrid training.

We respond to each reviewer in more detail below their respective comments.

Furthermore, in the process of expanding the derivation requested by Reviewer 8skU, we noticed a minor notational inconsistency in Eq. (4) of the current draft.
The bound is in fact derived for the Frobenius cosine $cos_F(J_\ell,\hat{J_\ell})=\frac{\langle J_\ell \hat{J_\ell}\rangle_F}{||J_\ell||_F||\hat{J_\ell}||_F},$  not for the signed average of per-column cosines. We will correct the notation accordingly in the revised version. This issue does not however affect the mathematical derivations, the algorithm or the experiments, only the way the alignment quantity is named in the text.

[1] Akrout, M., Wilson, C., Humphreys, P., Lillicrap, T. and Tweed, D.B., 2019. Deep learning without weight transport. Advances in neural information processing systems, 32.

[2] Roy, A. et al. (2025) 'Unlocking SVD-Space for Feedback Aligned Local Training'.

[3] Apolinario, M.P., Roy, A. and Roy, K., 2025, February. Lls: local learning rule for deep neural networks inspired by neural activity synchronization. In 2025 IEEE/CVF Winter Conference on Applications of Computer Vision (WACV) (pp. 7807-7816). IEEE.

[4] Nokland, A. and Eidnes, L.H., 2019, May. Training neural networks with local error signals. In International conference on machine learning (pp. 4839-4850). PMLR.

---

### Author Response · Authors · 2025-11-26
**Updated document : Revisions Summary**

We thank all reviewers for their detailed feedback, we updated the submitted pdf.
Below is a concise summary of the main changes, with pointers to the revised manuscript.

---

### 1. Forward-gradient theory and convergence

- **Frobenius cosine alignment bound.**
  We now measure alignment between the true Jacobian $\mathcal J_l$ and the JVP-based rank-1 estimator $\widehat{\mathcal J_l}$ using the *Frobenius* cosine and prove
  $$
  \mathbb{E}[\cos_F(\mathcal J_l,\widehat{\mathcal J_l})]
  \ge \sqrt{2/(\pi n_l)} \|\mathcal J_l\|_2/\|\mathcal J_l\|_F > 0,
  $$
  with an $O(1/\sqrt{B})$ concentration rate for the batched estimator.

  - **Sec. 3.2 (Eq. (4)) and App. B.1–B.3**.

- **Stochastic convergence theorem.**
  We added a standard stochastic-approximation theorem: under Lipschitz gradients, Robbins–Monro steps, and an explicit positive expected-cosine condition
  $\mathbb{E}[\langle g_t,d_t\rangle\mid\mathcal F_t]\ge \kappa\|g_t\|^2$,
  one obtains convergence to stationarity **in expectation**. The full proof is included.
   - **App. B.4**, with the role of Zoutendijk clarified in **Sec. 3.2 (“Zoutendijk theorem”)**.

- **“Compounding noise” through $\hat J_l$.**
  We added a Frobenius cosine composition lemma: using a Gram-matrix argument we lower-bound $\cos_F(B_l,J_l)$ in terms of $cos_F(B_l,\hat J_l)$ and $\cos_F(\hat J_l,J_l)$, clarifying how the two sources of noise interact.
  - **App. B.5**, referenced in **Sec. 3.2**.

- **Per-column vs. Frobenius cosine.**
  We explain when the empirical mean of per-column cosines is a good proxy for the Frobenius cosine used in the analysis (normalized columns and moderate variation of JVP column norms).
  - **App. C** and definition in **Sec. 3.2**.

---

### 2. Relation to prior work

- **Weight mirroring, FDFA, SVD-space methods.**
  We expanded **Sec. 3.1** to:
  - Present Akrout et al. (2019) as an important *precursor* showing learnable feedback weights in FA (but still sequential).
  - Discuss Kolen–Pollack / FDFA-style rules (Webster et al., Bacho & Chu) that track BP updates, and point to reproducibility issues we observed with FDFA.
    - **Details in App. A4** as before.
  - Contrast Roy et al. (2025), which optimizes a composite set of SVD-space local losses (including a cosine-like term), with GrAPE’s simpler JVP-based cosine loss and global alignment guarantee.

- **Forward-only local rules (no FA/DFA-style feedback).**
  At the end of **Sec. 2.2**, we added a paragraph on:
  - Nokland & Eidnes (2019): local classifiers and similarity losses attached to each layer.
  - Apolinario et al. (2025, LLS): local cross-entropy alignment with fixed periodic bases.
  We position GrAPE as complementary: it also uses forward information, but to learn feedback matrices for DFA-style parallel updates with explicit alignment guarantees.

---

### 3. Runtime, parallelism, and calibration cost

- **Serialized runtime overhead.**
  In our current Biotorch implementation (single GPU, largely serialized), GrAPE incurs a **6–20%** per-step wall-clock overhead vs. DFA/BP due to extra JVPs and alignment updates.
  - **Sec. 4.1**, first paragraph.

- **Layer-parallel Transformer prototype.**
  To illustrate potential gains once layer parallelism is exploited, we added preliminary timings for a small Transformer (hidden size 128, depths 2/4/8, batch 256, seq. length 64) using Python-level CUDA streams :
  BP: (9.1, 17.5, 35.8) ms vs. GrAPE: (3.0, 6.2, 12.1) ms.
  - **App. D.2, Table 5**, dedicated paragraph in **Sec. 4.1**.
  We label these as conservative, preliminary results (no kernel fusion), used only to illustrate the critical-path reduction story.

- **Calibration schedule and cost.**
  We clarified that calibration is **one BP step on a single mini-batch every $T$ epochs**, and quantified the amortized cost (e.g., $\approx 0.5$% of BP backward passes for ResNet-20/CIFAR-100 when $T=1$).
  - **Sec. 3.4**, **Sec. 4.1** ("cost of BP calibration), and correction of **Algorithm 1**.

---

### Meta-Review · Area_Chair_7sw2 · 2026-01-07

**Summary:**

This paper considers a variant of Feeback Alignment using rank 1 Jacobian estimates.

The main concerns of the reviewers were:
1. The validity of the theory stated in the submission
2. The practicality of the method (in particular, parallelisation, calibration)
3. The relation w.r.t to some prior work


I believe that 2. and 3. have been satisfactorily addressed by the authors. However, I am not convinced by the rebuttal regarding 1. The details of my metareview on the concerns and their rebuttal can be found in the next box.

Overall, given the heuristic nature of the formal statement, I strongly suggest that the authors lower their claims in the title, abstract, and introduction. The current formulation of the title, abstract, and contribution section is misleading and suggests that one of the paper's contributions is a formal convergence proof for GrAPE. For instance, in the contribution section:
> We derive a positive expected alignment bound for our rank-1 Jacobian estimator and a standard convergence-in-expectation result under a positive expected cosine condition

could be understood as a convergence in expectation result for GrAPE, but the "positive expected alignment bound for our rank-1 Jacobian estimator"  (4) does not seem to match the required positive expected cosine condition (14) to get such a convergence result.

For these reasons, I consider the paper borderline, but given the strong empirical results, I decided to put it above the acceptance threshold.

**Reviewer Concerns:**

The main concerns of the reviewers were:
1. The validity of the theory stated in the submission
2. The practicality of the method (in particular, parallelisation, calibration)
3. The relation w.r.t to some prior work


I believe that 2. and 3. have been satisfactorily addressed by the authors. However, I am not fully convinced by the rebuttal regarding 1. Here is the detail of the concerns and their rebuttal

### The convergence proof of the algorithm is heuristic rather than theoretical

I believe this concern has not been addressed by the authors. The assumption made in the proof of Theorem B.1 does not match the lower bound obtained for the algorithm (13). Moreover, the result provided in Theorem B.1 is standard, and thus proper references should be provided (or at least related work on previous results of convergence under positive expected cosine should be provided).

In my opinion, the challenge would be to either:
- show that (4) is sufficient for convergence (and thus it would require a modification of Theorem B.1)
- Show that (14) is true for GrAPE.

### “Compounding noise” and "expected positive cosine alignment"

The reviewers have addressed these two points. Though I am not sure whether these lower bounds are sufficient to obtain convergence. They are an important first step and justify a heuristically, why the method could converge.

Overall, I would suggest that the authors lower their claims in the title, abstract and introduction, which could be misleading:
> We derive a positive expected alignment bound for our rank-1 Jacobian estimator and a standard convergence-in-expectation result under a positive expected cosine condition

could be understood as a convergence in expectation result for GrAPE, but the "positive expected alignment bound for our rank-1 Jacobian estimator"  (4) does not seem to match the required positive expected cosine condition (14) to get such a convergence result.

**Reviewer Scores:**

Given that Reviewer DBs7 gave:
> First rigorous convergence argument for DFA-like methods based on Zoutendijk’s theorem.

and

> Reproducibility and rigor: transparent experimental setup, use of BioTorch, and fair baselines.

as strengths of this paper and that:
- I believe the convergence argument is only heuristic
- The reproducibility is limited as the code for the experiments is not provided (as noted by Reviewer AHBU)

I think Reviewer DBs7  would have considered lowering their score.

---

### Decision · Program_Chairs · 2026-01-26

Accept (Poster)